# Grammar-Induced Geometry for Data-Efficient Molecular Property Prediction

## Abstract

The prediction of molecular properties is a crucial task in the field of material and drug discovery. The potential benefits of using deep learning techniques are reflected in the wealth of recent literature. Still, these techniques are faced with a common challenge in practice: Labeled data are limited by the cost of manual extraction from literature and laborious experimentation. In this work, we propose a data-efficient property predictor by utilizing a learnable hierarchical molecular grammar that can generate molecules from grammar production rules. Such a grammar induces an explicit geometry of the space of molecular graphs, which provides an informative prior on molecular structural similarity. The property prediction is performed using graph neural diffusion over the grammar-induced geometry. On both small and large datasets, our evaluation shows that this approach outperforms a wide spectrum of baselines, including supervised and pre-trained graph neural networks. We include a detailed ablation study and further analysis of our solution, showing its effectiveness in cases with extremely limited data (only $\sim 100$ samples), and its extension to application in molecular generation.

## 1 Introduction

Molecular property prediction is an essential step in the discovery of novel materials and drugs, as it applies to both high-throughput screening and molecule optimization. Recent advances in machine learning, particularly deep learning, have made tremendous progress in predicting complex property values that are difficult to measure in reality due to the associated cost. Depending on the representation form of molecules, various methods have been proposed, including recurrent neural networks (RNN) for SMILES strings (Lusci et al., 2013; Goh et al., 2017), feed-forward networks (FFN) for molecule fingerprints (Tao et al., 2021b;a), and, more dominantly, graph neural networks (GNN) for molecule graphs (Yang et al., 2019; Bevilacqua et al., 2022; Aldeghi & Coley, 2022; Yu & Gao, 2022). They have been employed to predict biological and mechanical properties of both polymers and drug-like molecules. Typically, these methods learn a deep neural network that maps the molecular input into an embedding space, where molecules are represented as latent features and then transformed into property values. Despite their promising performance on common benchmarks, these deep learning-based approaches require a large amount of training data in order to be effective (Audus & de Pablo, 2017; Wieder et al., 2020).

In practice, however, scientists often have small datasets at their disposal, in which case deep learning fails, particularly in the context of polymers (Subramanian et al., 2016; Altae-Tran et al., 2017). For example, due to the difficulty of generating and acquiring data—which usually entails synthesis, wet-lab measurement, and mechanical testing— state-of-the-art works on polymer property prediction using real data are limited to only a few hundred samples (Menon et al., 2019; Chen et al., 2021). To compensate for the scarcity of experimental data, applied works often rely on labeled data generated by simulations, such as density functional theory and molecular dynamics (Aldeghi & Coley, 2022; Antoniuk et al., 2022). Yet, these techniques suffer from high computational costs, tedious parameter optimization, and a considerable discrepancy between simulations and experiments (Afzal et al., 2020; Chen et al., 2021), which limit their applicability in practice.

Recent deep learning research has recognized the scarcity of molecular data in several domains and has developed methods handling small datasets, including self-supervised learning (Zhang et al., 2021; Rong et al., 2020; Wang et al., 2022; Ross et al., 2021), transfer learning (Hu et al., 2020),

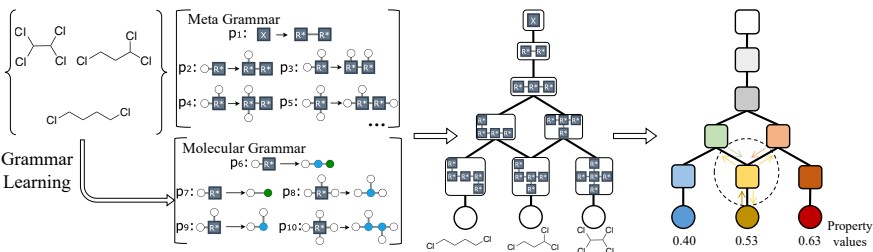

Figure 1: Overview. Given a set of molecules, we learn a hierarchical molecular grammar that can generate molecules from production rules. The hierarchical molecular grammar induces an explicit geometry for the space of molecules, where structurally similar molecules are closer in distance along the geometry. Such a grammar-induced geometry provides an informative prior for data-efficient property prediction. We achieve this by using graph neural diffusion over the geometry.

and few-shot learning (Guo et al., 2021b; Stanley et al., 2021). These methods involve pre-training networks on large molecular datasets before being applied to domain-specific, small-scale target datasets. However, when applied to datasets of very small size (e.g., ∼300), most of these methods are prone to perform poorly and are statistically unstable (Hu et al., 2020). Moreover, as we will show in our experiments, these methods are less reliable when deployed on target datasets that contain significant domain gaps from the pre-trained dataset (e.g., inconsistency in molecule sizes).

As an alternative to pure deep learning-based methods, formal grammars over molecular structures offer an explicit, explainable representation for molecules and have shown their great potential in addressing molecular tasks in a data-efficient manner (Kajino, 2019; Krenn et al., 2019; Guo et al., 2021a; 2022). A molecular grammar consists of a set of production rules that can be chained to generate molecules. The production rules, which can either be manually defined (Krenn et al., 2019; Guo et al., 2021a) or learned from data (Kajino, 2019; Guo et al., 2022), encode necessary constraints for generating valid molecular structures, such as valency restrictions. A molecular grammar has the combinatorial capacity to represent a large amount of molecules using a relatively small number of production rules. It has thus been adapted as a data-efficient generative model (Kajino, 2019; Guo et al., 2022). While molecular generation based on grammars has been widely studied and is relatively straightforward, extending the data-efficiency advantage of grammars to property prediction has not yet been well-explored.

**Motivation.** In this paper, we propose a framework for highly data-efficient property prediction based on a learnable molecular grammar. The intuition behind is that the production rule sequences for molecule generation provide rich information regarding the similarity of molecular structures. For instance, two molecular structures that share a common substructure would use the same sequence of grammar production rules. As it is widely recognized in cheminformatics that molecules with similar structures have similar properties (Johnson & Maggiora, 1990; Martin et al., 2002), grammar production sequences can thus be used as a strong structural prior to predict molecular properties. We aim to develop a model that explicitly represents grammar production sequences and captures the structure-level similarity between molecules. Even from a few molecules, this model is expected to reveal a wealth of information regarding the structural relationship between molecules, providing the key to a data efficient property predictor.

**Framework.** Figure 1 outlines our approach. At the heart of our method is a grammar-induced *geometry* (in the form of a graph) for the space of molecules. In the geometry, every path tracing from the root to a leaf represents a grammar production sequence that generates a particular molecule. Such a geometry explicitly captures the intrinsic closeness between molecules, where structurally similar molecules are closer in distance along the geometry. In contrast to the embedding space used in most deep learning methods, our grammar-induced geometry is entirely *explicit*, which can be integrated with the property predictor by considering all involved molecules simultaneously. To construct the geometry, we propose a *hierarchical molecular grammar* consisting of two parts: a pre-defined *meta grammar* at the top and a learnable *molecular grammar* at the bottom. We provide both theoretical and experimental evidence to demonstrate that the hierarchical molecular grammar is compact yet complete. To predict properties, we exploit the structural prior captured by the grammar-induced geometry using a graph neural diffusion model over the geometry. A joint optimization framework is proposed to simultaneously learn both the geometry and the diffusion.

**Contributions.** Our evaluation covers two small datasets of polymers (each containing ∼300 samples) and two large datasets of drug-like molecules (containing ∼600 and ∼1,000 samples, respectively). Our approach outperforms both competitive state-of-the-art GNN approaches and pretrained methods by large margins. A further analysis shows that when trained on only a subset of the training data (∼100 samples), our method can achieve performance comparable to pre-trained GNNs fine-tuned on the whole training set of the downstream prediction task, thus demonstrating the effectiveness of our method on extremely small datasets. We also show that our method can be integrated with molecular generation to guide the discovery of molecules with optimal properties.

**Related Work.** Our method is mainly related to three areas: 1) machine learning, particularly graph neural networks for molecular property prediction, in which we target the same problem but propose an entirely new approach; 2) grammars for molecular machine learning, which are used in existing works for molecular generation, while we use grammars to induce a geometry for molecular property prediction; and 3) geometric deep learning, where deep networks are applied to non-Euclidean domains. We briefly discuss the latter two in Section 2 as necessary background for our approach and refer to Appendix G for an in-depth discussion of related works.

## 2 PRELIMINARIES

**Molecular Hypergraph Grammar (MHG).** In MHGs, molecules are represented as hypergraphs $H = (V, E)$, where nodes represent chemical atoms and hyperedges represent chemical bonds or ring-like substructures. A MHG $G = (\mathcal{N}, \Sigma, \mathcal{P}, \mathcal{X})$ contains a set $\mathcal{N}$ of non-terminal nodes, a set $\Sigma$ of terminal nodes representing chemical atoms, and a starting node $\mathcal{X}$. It describes how molecular hypergraphs are generated using a set of production rules $\mathcal{P} = \{p_i | i = 1, ..., k\}$ of form $p_i : LHS \rightarrow RHS$, where the left- and right-hand sides ($LHS$ and $RHS$) are hypergraphs. Starting at $\mathcal{X}$, a molecule is generated by iteratively selecting a rule whose $LHS$ matches a sub-hypergraph in the current hypergraph and replacing it with $RHS$ until only terminal nodes remain. For each production rule, the $LHS$ contains only non-terminal nodes and no terminal nodes, whereas the $RHS$ can contain both non-terminal and terminal nodes. Both sides of the production rule have the same number of anchor nodes, which indicate correspondences when $LHS$ is replaced by $RHS$ in a production step. For a formal definition, see Guo et al. (2022).

**Graph Diffusion** models the information propagation between nodes on a graph using heat equations. The node features are updated following a diffusion PDE as follows:

$$\mathbf{U}_T = \mathbf{U}_0 + \int_0^T \frac{\partial \mathbf{U}_t}{\partial t} \mathrm{d}t, \qquad \frac{\partial \mathbf{U}_t}{\partial t} = \mathbf{L}_\alpha \mathbf{U}_t, \qquad (1)$$

where matrix $\mathbf{U}_t$ represents the features of all nodes in the graph at time $t$ and the matrix $\mathbf{L}_\alpha$, which has the same sparsity structure as a graph Laplacian, represents the diffusivity defined on all edges in the graph. $\mathbf{L}_\alpha$ is calculated using $a(\cdot, \cdot; \alpha)$ parameterized by $\alpha$, i.e. $L_{ij} = a(U_t^{(i)}, U_t^{(j)}; \alpha)$ for all connected node pairs $(i, j)$. For more details, see Chamberlain et al. (2021).

**Notation.** For a graph grammar $G = (\mathcal{N}, \Sigma, \mathcal{P}, \mathcal{X})$, we say a graph $H$ can be *derived* from the grammar if there is a sequence of production rules from $\mathcal{P}$ that generates this graph, denoted by $\mathcal{X} \overset{*}{\underset{\mathcal{P}}{\Rightarrow}} H$. $H_1 \overset{p}{\underset{\mathcal{P}}{\Rightarrow}} H_2$ denotes one-step grammar production that transforms graph $H_1$ to graph $H_2$ using rule $p$. As a special form of general graph, a tree is denoted as $T$ and the set of all possible trees is denoted as $\mathcal{T}$. Note that all trees discussed in this paper are *unrooted without order*, i.e. connected acyclic undirected graphs (Bender & Williamson, 2010). $\Delta(T)$ denotes the maximal degree of $T$. $TED(T_1, T_2)$ denotes the *tree edit distance* between tree $T_1$ and tree $T_2$ as defined in Zhang (1996) and Paaßen (2018). $|\mathcal{P}|$ denotes the number of rules in a production rule set $\mathcal{P}$ and $|T|$ denotes the size of a tree $T$.

## 3 GRAMMAR-INDUCED GEOMETRY FOR PROPERTY PREDICTION

### 3.1 GRAMMAR-INDUCED GEOMETRY

**Problem Formulation.** A molecular property predictor can be expressed as a function $\pi(\cdot) : \mathcal{H} \rightarrow \mathbb{R}$ that maps molecules formulated as hypergraphs $H = (V, E) \in \mathcal{H}$ into scalar values. The mapping

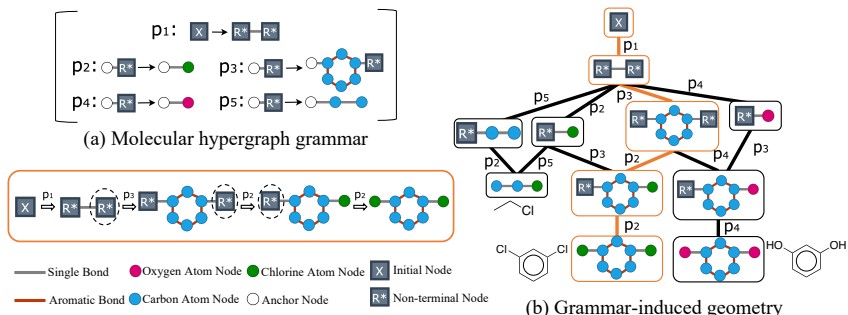

(a) Molecular hypergraph grammar

(b) Grammar-induced geometry

Figure 2: (a) Example of a molecular hypergraph grammar and (b) its induced geometry. All possible production sequences to generate three molecules are outlined in the geometry, where the nodes represent intermediate or molecular hypergraphs and the edges represent production steps. Structurally similar molecules have a smaller shortest path between them and are thus closer in distance along the geometry.

function $\pi = g \circ f$ contains two components: an embedding function $f(\cdot) : \mathcal{H} \to \mathbb{R}^n$ that maps input hypergraphs into an Euclidean *latent feature space* (also known as the *embedding space*), and a transformation function $g(\cdot) : \mathbb{R}^n \to \mathbb{R}$ (usually a simple linear function) which maps latent features into property values. The embedding function is designed to capture the similarity between molecules, such that molecules with similar properties are closer in the embedding space in terms of Euclidean distance (Cayton, 2005). The key reason why most supervised machine learning methods fail in data-sparse cases is that the embedded function fails to capture molecular similarity since it only has access to a limited number of samples with property labels. To address this issue, we propose an additional, explicit prior on molecular similarity that does not depend on data labels, but provides valuable information for property prediction.

**Grammar-induced Geometry.** Our efforts are motivated by the fact that *structure-level similarity* between molecules is known to be strongly correlated with their properties, e.g., organic molecules sharing a functional group are proven to have similar chemical properties (Johnson & Maggiora, 1990; Martin et al., 2002). We *explicitly* model this structure-level similarity as an informative prior for data-efficient property prediction.

Inspired by the fact that grammars are generative models for molecules based on combinations of molecular substructures, we exploit the molecular hypergraph grammar to capture the structural similarity between molecules. Figure 2 outlines how a molecular hypergraph grammar $G = (\mathcal{N}, \Sigma, \mathcal{P}, \mathcal{X})$ can be used to construct a *geometry* for the space of molecules in the form of a graph $\mathcal{G} = (\mathcal{V}, \mathcal{E})^1$. The geometry enumerates all possible production sequences, where every possible intermediate or molecular hypergraph is represented as a node in $\mathcal{V} = \{v | v = H_v(V_v, E_v)\}$, and each production step is represented as an edge in $\mathcal{E} = \{(s, t) | H_s \overset{p}{\Rightarrow} H_t, p \in \mathcal{P}\}$. In this geometry, every leaf node represents a molecule. Every path tracing from the root $H_{root} = (\mathcal{X}, \varnothing)$ to a leaf represents a production sequence that generates the corresponding molecule. Since all the molecules can be derived from the initial node $\mathcal{X}$, any two leaves representing two molecules are connected in this geometry, where the path between them represents the sequence of transforming from one molecular structure to the other. More structurally similar molecules have a smaller *shortest path* as their molecular hypergraphs share a common intermediate hypergraph. The distance between two molecules is defined as the shortest-path distance between them along the geometry with unit weight at each edge. As a result, structurally similar molecules are closer in distance along the geometry. We use this geometry as an additional input to the molecular property predictor: $\pi'(\cdot, \cdot) : \mathcal{H} \times \mathfrak{G} \to \mathbb{R}$, where $\mathcal{G} \in \mathfrak{G}$ is the grammar-induced geometry. The geometry can be optimized in conjunction with the property predictor in order to minimize the prediction loss. As the geometry is determined by the grammar, the optimization of the geometry can be converted into the learning of production rules, where the latter can be achieved using the method described in Guo et al. (2022).

The crucial remaining question is how to construct this geometry $\mathcal{G} = (\mathcal{V}, \mathcal{E})$ from a given molecular hypergraph grammar? A key characteristic of the grammar-induced geometry is that each node in

---

[1]To distinguish the graph of the geometry from the graph of molecules, we use *hypergraphs* to formulate the molecular graphs generated from grammar production.

the geometry represents a *unique* intermediate hypergraph or molecular hypergraph. This ensures that the geometry includes the minimal path between two molecules. To satisfy this characteristic, we use breadth-first search (BFS) originating from the root $H_{root} = (\mathcal{X}, \varnothing)$ to iteratively expand the geometry. At each iteration when we visit a node $v$, we enumerate all the production rules that could be applied to the intermediate hypergraph $H_v$ represented by the current node. Then we produce a new hypergraph $H_{new}^{(i)}$ for each applicable rule $p_i$: $\{H_{new}^{(i)} | H_v \xrightarrow{p_i} H_{new}^{(i)}, p_i \in \mathcal{P}\}$. For each new hypergraph $H_{new}^{(i)}$, if $H_{new}^{(i)} \neq H_{v'}$ for all $v' \in \mathcal{V}$, we create a new node $v_{new}^{(i)} = H_{new}^{(i)}$, add it to the node set $\mathcal{V}$, and create a new edge $(v, v_{new}^{(i)})$ that gets added to the edge set $\mathcal{E}$ of the geometry. If there is an isomorphic hypergraph in the existing node set, i.e. $H_{\hat{v}} = H_{new}^{(i)}, \hat{v} \in \mathcal{V}$, we do not create any new node; instead, we add an edge $(v, \hat{v})$ between the matched node $\hat{v}$ and the currently visited node $v$. The algorithm terminates once all the molecules involved in the property prediction task have been visited in the geometry.

In practice, the construction of grammar-induced geometry is very costly, and often computationally intractable for more complex grammars. By testing with random grammars learned from Guo et al. (2022), we find that it is infeasible to construct the geometry when there are more than ten production rules. The major bottleneck comes from the combinatorial complexity of production rules: The more rules a grammar has, the more intermediate hypergraphs it can generate. As the depth of the geometry increases, the number of intermediate hypergraphs increases exponentially. This incurs a significant computational cost, as each newly expanded hypergraph requires pair-wise isomorphism tests. This cost also poses a serious obstacle to the optimization of the geometry, since the geometry must be rebuilt at each optimization iteration.

## 3.2 Hierarchical Molecular Grammars

We propose a *hierarchical molecular grammar* to address the computational challenge mentioned above. Our insight stems from the fact that every hypergraph can be represented as a tree-structure object, i.e. a *junction tree* using tree decomposition (Diestel, 2005). The tree representation brings two benefits: 1) It allows us to measure the underlying structural similarity between molecules by reflecting the scaffold of the substructure components (Jin et al., 2018); and 2) as a more compact representation, the possible junction tree structures are considerably fewer than molecular hypergraphs, and therefore can be enumerated. By carefully designing the form of grammar rules, we divide the generation process of a molecular hypergraph into two parts: first generating a tree with homogeneous tree nodes, then converting the tree into a molecular hypergraph by specifying a sub-hypergraph per tree node. The grammar-induced geometry can thus be divided into two parts: a top part where geometry nodes represent only trees and a bottom part where the leaves of the geometry represent molecular hypergraphs. Due to this hierarchical decomposition, the top part of the geometry is data-independent and can be computed offline to apply to any molecular datasets. At run-time, we only need to construct the bottom part of the geometry, which is formulated as a byproduct of grammar learning in our approach. As a result of the geometry construction, each molecule is connected to one junction tree in the top part. Molecular structure similarity is determined by the distance between their corresponding junction trees along the geometry.

A hierarchical molecular grammar consists of two sub-grammars that generate two parts of the geometry respectively. We refer to the sub-grammar that only generates trees as a *meta grammar* and the top-part geometry it constructs as a *meta geometry*. We provide formal definitions of meta grammar and demonstrate that despite the additional hierarchical constraint, our hierarchical molecular grammar is *as expressive as* the general molecular hypergraph grammar in Guo et al. (2022). Figure 3 shows an overview of a hierarchical molecular grammar and its induced geometry.

**Meta Grammars.** The formal definition of a meta grammar is as follows.

**Definition 1.** *A meta grammar $\overline{G} = (\mathcal{N}, \varnothing, \mathcal{P}_{\overline{G}}, \mathcal{X})$ is a hypergraph grammar, which only contains non-terminal nodes and only generates trees, i.e. $\forall w \in \{w | \mathcal{X} \underset{\mathcal{P}_{\overline{G}}}{\overset{*}{\Rightarrow}} w\}, w \in \mathcal{T}$.*

*A meta grammar $\overline{G}$ is $k$-degree if, for all trees $T$ of maximal degree $\Delta(T) \leq k$, we have $\mathcal{X} \underset{\mathcal{P}_{\overline{G}}}{\overset{*}{\Rightarrow}} T$.*

*A meta grammar $\overline{G}$ is edit-complete if, for any tree pair $(T, T')$ with $|T| < |T'|$, and tree edit distance $TED(T, T') = 1$, and $\mathcal{X} \underset{\mathcal{P}_{\overline{G}}}{\overset{*}{\Rightarrow}} T, T'$, there is a rule $p \in \mathcal{P}_{\overline{G}}$ such that $T \overset{p}{\Rightarrow} T'$.*

*A $k$-degree, edit-complete meta grammar $\overline{G}$ is minimal if there is no other such meta grammar $\overline{G}'$ with $|\mathcal{P}_{\overline{G}'}| < |\mathcal{P}_{\overline{G}}|$.*

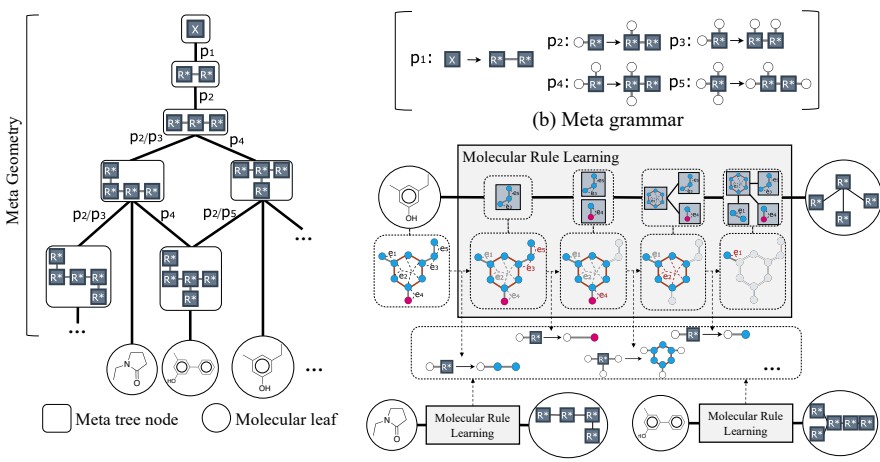

(a) Geometry induced by hierarchical molecular grammar  (c) Molecular grammar

Figure 3: (a) Overview of the geometry induced by hierarchical molecular grammar. (b) A pre-defined meta grammar is used to pre-compute the meta geometry offline. (c) At run-time, molecular grammar is obtained using molecular rule learning. Each molecule is converted into a junction tree and is connected to a meta tree node of the meta geometry that represents an isomorphic meta tree.

In Appendix H, we provide the formal construction of meta grammars for generating trees with arbitrary degree and more elaboration on the three additional attributes.

**Proposition 1.** *A meta grammar $\overline{G}$ with a meta rule set $\mathcal{P}_{\overline{G}}$ constructed in Appendix H is of degree $k$, edit-complete, and minimal.*

Appendix H also provides the proof that the three attributes are satisfied. Generally speaking, we construct meta rule sets of arbitrary degrees by induction from 1-degree meta grammar which consists of only one meta rule. In practice, a meta grammar with a reasonable degree should be chosen to ensure the coverage of most molecules. In our experiments, we find it sufficient to use a meta rule set of degree 4, which contains 8 rules in total.

**Hierarchical Molecular Grammars.** We define hierarchical molecular grammar as follows,

**Proposition 2.** *A* hierarchical molecular grammar $\mathbf{G} = (\overline{G}, G_{mol})$ *consisting of two hypergraph grammars: 1) a $k$-degree, edit-complete, and minimal meta grammar $\overline{G}$ defined in Definition 1; and 2) a* molecular grammar $G_{mol} = (\mathcal{N}, \Sigma, \mathcal{P}_{mol}, \varnothing)$ *where the RHS of any molecular rule in $\mathcal{P}_{mol}$ only contains terminal nodes (i.e. atoms), is a* complete *grammar.*

More discussion on the proposition is in Appendix H. The completeness of our hierarchical molecular grammar also shows that there is no loss of representation capacity as compared to general molecular hypergraph grammars.

**Geometry Construction and Molecular Grammar Learning.** The geometry induced by the hierarchical molecular grammar is constructed in two phases, as shown in Figure 3(a): the top for the meta grammar and the bottom for the molecular grammar. Since the meta grammar is pre-defined, the top meta geometry can be pre-computed offline following the BFS procedure described in Section 3.1 with a given maximum BFS depth. Each node in the meta geometry represents a tree of non-terminal nodes. We call the tree generated using the meta grammar a *meta tree* and the node in the meta geometry a *meta tree node*. We find it sufficient to use a maximum depth of 10 in practice. The bottom part of the geometry determines how each molecule is connected to the meta geometry as a *molecular leaf* through its junction tree structure. We obtain the junction trees of molecules simultaneously with the learning of molecular grammar rules. Figure 3(c) illustrates the process. Thus, the geometry optimization can be achieved by learning the molecular grammar.

We follow the grammar learning from Guo et al. (2022) but constrain the learned rules to contain only terminal nodes on the *RHS*, so as to satisfy the molecular grammar. Each molecule is considered as a hypergraph, and a set of hyperedges is iteratively sampled from the hypergraph until all hyperedges are visited. We construct the molecular rules along with the junction tree. At each iteration, a molecular rule is constructed for each connected component, where the *LHS* contains

a single non-terminal node and the $RHS$ contains the connected component. For the junction tree, we add a tree node representing the connected component and create a tree edge between two nodes if their corresponding connected components share a common hypergraph node. The iterative sampling follows an i.i.d. Bernoulli distribution based on a probability function $\phi(e; \theta)$ defined on each hyperedge $e$ with learnable parameters $\theta$. At the conclusion of the sampling process, we can obtain a junction tree of the molecular hypergraph along with a set of molecular rules for converting this junction tree back to the original hypergraph. The molecule is then connected to the meta tree node in the meta geometry that represents the isomorphic meta tree to the junction tree. We provide a detailed formulation of the constructed molecular rule and the resulting junction tree in Appendix I, and more details of the grammar-induced geometry construction in Appendix J.

### 3.3 Graph Diffusion over Grammar-induced Geometry

Property prediction requires a model that can operate on our grammar-induced geometry and is suitable for scarce data. We choose the graph neural diffusion model GRAND (Chamberlain et al., 2021) for its effectiveness in overcoming the oversmoothing that plagues most GNNs. Three learnable components are used in a graph diffusion process: an encoder function $\varphi$ defined on all the nodes in the grammar-induced geometry, a decoder function $\psi$ defined on molecular leaves, and a graph diffusion process given in Equation 1. Specifically, the encoder $\varphi$ yields the initial state of the diffusion process $\mathbf{U}_0 = \varphi(\mathbf{U}_{\text{in}})$, where $\mathbf{U}_{\text{in}}$ is the matrix form of the input features of all the nodes. The decoder produces predicted property values of all molecular leaves $\mathbf{u}_T = \psi(M \odot \mathbf{U}_T)$, where $\mathbf{U}_T$ is the node-feature matrix from the final diffusion state, $M$ is a binary mask that masks out the rows corresponding to non-molecule nodes, $\odot$ is the Hadamard product, and $\mathbf{u}_T$ is the resulting vector containing property values of all molecular leaves. Our optimization objective encodes the learning for both the grammar-induced geometry and the diffusion model:

$$\min_{\theta, (\varphi, \psi, \alpha)} l(\mathbf{u}_T, \hat{\mathbf{u}}) = \min_{\theta} \min_{(\varphi, \psi, \alpha)} l(\mathbf{u}_T, \hat{\mathbf{u}}), \tag{2}$$

where $\hat{\mathbf{u}}$ represents the vector of ground-truth property values for all leaves and $l(\cdot, \cdot)$ is a regression loss. Recall that the geometry is only determined by the molecular rules, so the molecular grammar learning parameters $\theta$ are the only parameters relevant for obtaining the geometry. Since $\theta$ and $(\varphi, \psi, \alpha)$ are two groups of independent variables, we exploit block coordinate descent (Wright, 2015) to optimize the objective. A detailed derivation is provided in Appendix K.

## 4 Evaluation

The experiments demonstrate the **generality of our approach** and answer the following questions:

- Does our approach outperform existing methods on **small datasets**?
- How well does our approach perform on **large, widely-studied datasets**?
- To what extent is our approach effective on **extremely small datasets**?
- Can our approach **guide generative models** toward molecules with optimal properties?

### 4.1 Experiment Setup

**Data.** We evaluate our approach on eight datasets: CROW (a curated dataset from literature), Permeability (Yuan et al., 2021), FreeSolv (Mobley & Guthrie, 2014), Lipophilicity (Wang et al., 2015), HOPV (Lopez et al., 2016), DILI (Ma et al., 2020), and PTC (Xu et al., 2018). These datasets cover: 1) commonly used benchmark datasets including MolecuNet (FreeSolv, Lipophilicity, HOPV, ClinTox) and TUDataset (PTC), 2) both classification (DILI, PTC, ClinTox) and regression tasks (CROW, Permeability, FreeSolv, Lipophilicity, HOPV), and 3) sizes that are small (CROW, Permeability, HOPV, PTC) and large (FreeSolv, Lipophilicity, ClinTox). We report the mean absolute error (MAE) and the coefficient of determination ($R^2$) for regression, and Accuracy and ROC-AUC scores for classification. See Appendix E for more details.

**Baselines.** We compare our approach with various approaches: Random Forest, FNN, wD-MPNN (D-MPNN), ESAN, HM-GNN, PN, and Pre-trained GIN. For descriptions, see Appendix F. To show the generalizability of our pipeline (called Geo-DEG), we implement two versions, each with a different diffusion encoder, GIN and MPNN. Appendix F provides the implementation details.

Table 1: Results on two small datasets of polymers (best result **bolded**, second-best underlined). Our approach (Geo-DEG) outperforms state-of-the-art GNNs and pre-trained methods by large margins.

| | CROW | | Permeability | |
|---|---|---|---|---|
| | **MAE** ↓ | **$R^2$** ↑ | **MAE** ↓ | **$R^2$** ↑ |
| Random Forest | 27.9 ± 3.2 | 0.67 ± 0.08 | 0.58 ± 0.01 | 0.72 ± 0.03 |
| FFN | 24.0 ± 2.1 | 0.84 ± 0.02 | 0.56 ± 0.04 | 0.73 ± 0.06 |
| wD-MPNN | 20.6 ± 1.3 | 0.89 ± 0.02 | 0.46 ± 0.03 | 0.80 ± 0.03 |
| ESAN | 26.1 ± 1.3 | 0.79 ± 0.02 | 0.40 ± 0.03 | 0.81 ± 0.03 |
| HM-GNN | 30.8 ± 1.8 | 0.76 ± 0.01 | 0.49 ± 0.03 | 0.68 ± 0.01 |
| PN (finetued) | 21.1 ± 1.3 | 0.89 ± 0.01 | 0.48 ± 0.04 | 0.70 ± 0.03 |
| Pre-trained GIN (finetued) | 19.3 ± 1.4 | 0.91 ± 0.01 | 0.46 ± 0.04 | 0.69 ± 0.02 |
| **Geo-DEG (GIN)** | **17.0 ± 1.4** | **0.92 ± 0.01** | 0.34 ± 0.02 | **0.84 ± 0.02** |
| **Geo-DEG (MPNN)** | 18.5 ± 1.2 | 0.91 ± 0.01 | **0.32 ± 0.03** | 0.83 ± 0.02 |

## 4.2 RESULTS ON SMALL DATASETS

To answer the first question posted at the beginning, we conduct experiments on five small datasets: CROW, Permeability, HOPV, DILI, and PTC. Table 1 shows the results on the first two datasets, which contain polymers. Regarding MAE, the baselines ESAN and HM-GNN are only comparable with simple, non-deep learning methods (e.g., random forest), while pre-trained GIN and wD-MPNN perform reasonably well. Both variants of our method outperform all the other methods by a large margin. Results and discussion on the other datasets are given in Appendix A.

**Discussions.** In general, CROW is a more challenging dataset than Permeability, as it has fewer samples and a broader range of property values (with a standard derivation of 86.8 versus 1.24 for Permeability). On CROW, traditional machine learning methods such as Random Forest and FNN are quite competitive and even outperform modern GNN architectures such as ESAN and HM-GNN. wD-MPNN achieves reasonable performance thanks to the special graph representation for polymer ensembles. PN and Pre-trained GIN perform exceptionally well on CROW due to their pre-training on large datasets. However, this pre-training does not help on Permeability, which has a much larger average molecule size (with an average molecular weight of 391.8 versus 153.8 for CROW). Thus, there is a domain gap between the dataset where PN and Pre-trained GIN are pre-trained and the Permeability dataset, resulting in poor performance of both methods. ESAN benefits from the subgraph selection scheme on large molecular graphs and performs well on Permeability. The poor performance of HM-GNN on both datasets shows that it is not as effective in regression tasks as it is in classification tasks. The overall results show that: 1) our method is capable of handling polymers with varying molecular sizes, and 2) the superior performance of our method when coupled with either GIN or MPNN confirms its generalizability to different feature extractors.

## 4.3 RESULTS ON LARGE DATASETS

To answer the second question, we conduct experiments on three large datasets: FreeSolv, Lipophilicity, and ClinTox. Tables 2 and 5 in Appendix A show the results. For FreeSolv and Lipophilicity, simple machine learning methods including random forest and FFN perform poorly. Pre-trained GNNs cannot perform as well as they do on small polymer datasets. The performance of advanced GNN architectures, especially D-MPNN, is reasonably good on both datasets. Our method equipped with MPNN diffusion encoder performs the best among all approaches. Results and discussion on the other datasets are given in Appendix A.

**Discussions.** On large datasets, random forest and simple FFN become less competitive than on small datasets. This is reasonable since larger datasets require larger model capacities to represent the relationship between molecules and their properties. Pre-training does not significantly help on either dataset due to the inherent domain gap between datasets used for pre-training versus those used for fine-tuning. Among the three GNN-based baselines, D-MPNN performs the best and exhibits better stability than ESAN and HM-GNN. Our approach can further improve the performance of D-MPNN and outperforms all the baselines. These results demonstrate that our method is scalable and competitive with state-of-the-art methods for large molecule datasets.

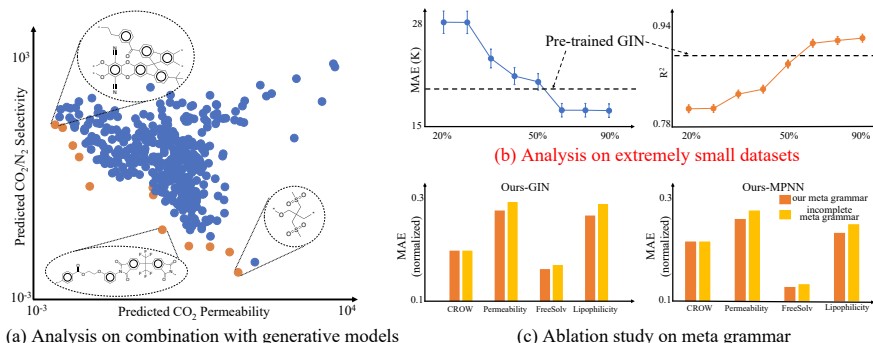

(a) Analysis on combination with generative models     (c) Ablation study on meta grammar

Figure 4: (a) Two objectives from the Permeability dataset predicted using our method combined with the grammar-based generative model. Molecules lying on the Pareto front are highlighted. (b) Performance of our method trained on subsets of CROW training data with different ratios. Even when the training set is halved, our approach can still achieve competitive results compared to Pre-trained GIN fine-tuned on the whole training set. (c) Comparison of performance using different meta grammars, demonstrating the necessity of a complete meta grammar.

## 4.4 ANALYSIS

This section examines the effectiveness of our method on extremely small datasets, its combination with generative models that answer the last two questions posted at the beginning, and an ablation study on the meta grammar.

**Performance on Extremely Small Datasets.** We conduct a study on the minimum viable number of samples for our method. We randomly sample $20\%$ of CROW as a fixed testing set. Using the remaining data, we construct eight randomly selected training sets, each consisting of a portion of all remaining data, ranging from $20\%$ to $100\%$. These eight training sets are used to train our method using GIN as the feature extractor. Figure 4(b) shows the performance on the testing set. Even when the training set is halved (with only $94$ samples), our approach still achieves results that are comparable to those of the Pre-trained GIN fine-tuned on the whole training set. Appendix B includes a more detailed study for the effect of changing dataset training size on model performance.

**Combination with Generative Models.** In our optimization framework, grammar metrics can be considered as additional objectives, allowing us to jointly optimize both generative models and property predictors. Following Guo et al. (2022), we use diversity and Retro* score as our grammar metrics and then perform joint optimization on the Permeability dataset. After training, we generate $400$ molecules and predict their property values using our approach, including all six property types from Permeability. Figure 4(a) shows two out of six predicted property values with the Pareto front highlighted. Clearly, our approach can be combined with generative models to provide a comprehensive pipeline for the discovery of optimal molecules. The use of Retro* also enables finding synthesis paths of generated molecules as shown in Appendix L.

**Ablation Study on Meta Grammar.** In this study, we examine the necessity of the meta rules in our hierarchical grammar. We remove two rules that have degree 4 on the $LHS$ from the 4-degree meta grammar, resulting in a meta geometry with same number of nodes but $10\%$ fewer edges than the one used in our main experiments. With this modified meta grammar, we run the pipeline for all four datasets and compare the results with the original meta grammar in Figure 4(c). All four datasets exhibit a performance drop when using the modified meta grammar. The results provide experimental evidence for the necessity of a complete meta grammar.

## 5 CONCLUSION

We propose a data-efficient molecular property predictor based on a hierarchical molecular grammar. The grammar induces an explicit geometry describing the space of molecular graphs, such that a graph neural diffusion on the geometry can be used to effectively predict property values of molecules on small training datasets. One avenue of future work is to extend our pipeline to model 3D molecular structures and to address general graph design problems.

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

## A    ADDITIONAL RESULTS

Table 2: Results on FreeSolv and Lipophilicity (best **bolded**, second-best underlined).

|  | FreeSolv | | Lipophilicity | |
|---|---|---|---|---|
|  | **MAE** ↓ | **R²** ↑ | **MAE** ↓ | **R²** ↑ |
| Random Forest | 4.58 ± 0.34 | 0.60 ± 0.09 | 0.64 ± 0.04 | 0.65 ± 0.03 |
| FFN | 3.67 ± 0.40 | 0.77 ± 0.05 | 0.51 ± 0.02 | 0.80 ± 0.03 |
| D-MPNN | 0.54 ± 0.08 | 0.90 ± 0.02 | 0.44 ± 0.02 | 0.90 ± 0.02 |
| ESAN | 0.67 ± 0.07 | 0.86 ± 0.03 | 0.46 ± 0.04 | 0.90 ± 0.01 |
| HM-GNN | 0.63 ± 0.05 | 0.86 ± 0.02 | 0.58 ± 0.04 | 0.84 ± 0.02 |
| PN (finetued) | 0.65 ± 0.04 | 0.85 ± 0.02 | 0.56 ± 0.04 | 0.81 ± 0.02 |
| Pre-trained GIN (finetued) | 1.01 ± 0.11 | 0.74 ± 0.03 | 0.52 ± 0.03 | 0.89 ± 0.01 |
| **Geo-DEG (GIN)** | 0.62 ± 0.06 | 0.90 ± 0.02 | 0.48 ± 0.02 | 0.88 ± 0.02 |
| **Geo-DEG (MPNN)** | **0.49 ± 0.06** | **0.94 ± 0.02** | **0.42 ± 0.02** | **0.91 ± 0.02** |

Table 3: Results on HOPV (best **bolded**, second-best underlined).

| **Methods** | **MAE (normalized)** ↓ | **R²** ↑ |
|---|---|---|
| Random Forest | 0.36 ± 0.03 | 0.69 ± 0.05 |
| FFN | 0.35 ± 0.03 | 0.67 ± 0.06 |
| D-MPNN | 0.36 ± 0.03 | 0.69 ± 0.04 |
| ESAN | 0.37 ± 0.02 | 0.66 ± 0.06 |
| HM-GNN | 0.40 ± 0.02 | 0.65 ± 0.05 |
| PN (finetued) | 0.42 ± 0.02 | 0.65 ± 0.04 |
| Pre-trained GIN (finetued) | 0.38 ± 0.02 | 0.66 ± 0.03 |
| **Geo-DEG (GIN)** | 0.32 ± 0.03 | 0.70 ± 0.03 |
| **Geo-DEG (MPNN)** | **0.30 ± 0.02** | **0.74 ± 0.03** |

Table 4: Results on DILI and PTC (best **bolded**, second-best underlined).

|  | DILI | | PTC | |
|---|---|---|---|---|
|  | **Accuracy** ↑ | **ROC-AUC** ↑ | **Accuracy** ↑ | **ROC-AUC** ↑ |
| Random Forest | 0.70 ± 0.09 | 0.80 ± 0.06 | 0.60 ± 0.06 | 0.63 ± 0.05 |
| FFN | 0.69 ± 0.07 | 0.78 ± 0.07 | 0.58 ± 0.06 | 0.61 ± 0.04 |
| D-MPNN | 0.75 ± 0.10 | 0.83 ± 0.07 | 0.67 ± 0.06 | 0.70 ± 0.05 |
| ESAN | 0.74 ± 0.10 | 0.82 ± 0.10 | 0.64 ± 0.08 | 0.68 ± 0.06 |
| HM-GNN | 0.76 ± 0.08 | 0.83 ± 0.09 | 0.66 ± 0.07 | 0.69 ± 0.06 |
| PN (finetued) | 0.75 ± 0.10 | 0.83 ± 0.06 | 0.61 ± 0.08 | 0.65 ± 0.07 |
| Pre-trained GIN (finetued) | 0.74 ± 0.09 | 0.82 ± 0.06 | 0.62 ± 0.09 | 0.66 ± 0.07 |
| **Geo-DEG (GIN)** | 0.76 ± 0.09 | 0.84 ± 0.05 | 0.64 ± 0.09 | 0.68 ± 0.06 |
| **Geo-DEG (MPNN)** | **0.78 ± 0.08** | **0.86 ± 0.06** | **0.69 ± 0.07** | **0.71 ± 0.07** |

**Results & Discussion.** For HOPV, since it is a small regression dataset, traditional methods (random forest and FFN) achieve competitive performance and are even better than several graph neural networks including ESAN and HM-GNN. Both variants of our method outperform other baselines by a large margin, which aligns with the results in the other two small regression datasets. In the two small classification datasets, DILI and PTC, there is no significant difference between baseline models regarding performance. The reason for this is that the test dataset is small, and the accuracy of the model can be limited by a few hard examples that cannot be classified correctly. It is therefore more informative to use the ROC-AUC score. Among the baselines, HM-GNN performs the best due to its motif-based representation. Our method can still outperform all other methods regarding both accuracy and ROC-AUC score. For ClinTox in Table 5, we compare our approach with a wide range of highly performant baseline methods from a recent paper Zhou et al. (2022), which

Table 5: Results on ClinTox (best **bolded**, second-best underlined).

| Methods | ROC-AUC ↑ |
|---|---|
| D-MPNN | 90.6 ± 0.6 |
| AttentiveFP | 84.7 ± 0.3 |
| N-Gram$_{RF}$ | 77.5 ± 4.0 |
| N-Gram$_{XGB}$ | 87.5 ± 2.7 |
| GROVER$_{base}$ | 81.2 ± 3.0 |
| GROVER$_{large}$ | 76.2 ± 3.7 |
| GraphMVP | 79.1 ± 2.8 |
| MolCLR | 91.2 ± 3.5 |
| GEM | 90.1 ± 1.3 |
| Pre-trained GIN | 72.6 ± 1.5 |
| Uni-Mol | 91.9 ± 1.8 |
| **Geo-DEG (GIN)** | 74.4 ± 1.8 |
| **Geo-DEG (MPNN)** | **92.2 ± 0.7** |

by itself proposes one of the SOTA methods on ClinTox. Note that these are very strong baseline methods, including many methods pre-trained using 3D molecular data. The overall results show that our method achieves significantly better performance on challenging small regression datasets and outperforms a wide spectrum of baselines on various common benchmarks.

## B   ANALYSIS ON TRAINING DATASET SIZE

We further extend our study in Section 4.4 for the effect of changing dataset training size on model performance by including the analysis of all three small regression datasets with more baselines compared. Figure 5 illustrates the results of Pre-trained GIN, wD-MPNN/D-MPNN, and two variants of our model on CROW, Permeability, and HOPV. We report the performance of each model trained using different numbers of training samples, randomly sampled from the original training dataset. Across different training dataset sizes, our proposed method consistently outperforms other baselines. A smaller training dataset leads to a larger performance improvement, demonstrating the data-efficiency of our model.

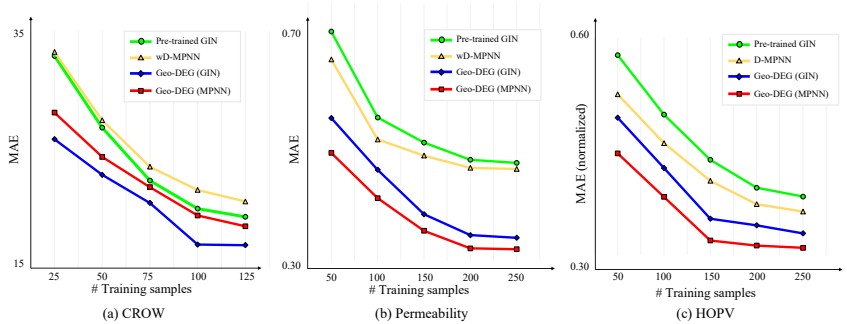

(a) CROW          (b) Permeability          (c) HOPV

Figure 5: Analysis on training dataset size.

## C   ANALYSIS ON CONSTRUCTION COST REDUCTION OF HIERARCHICAL MOLECULAR GRAMMAR

In this section, we provide empirical evidence on the construction cost reduction of our hierarchical molecular grammar compared with non-hierarchical molecular grammar from Guo et al. (2022). We conduct ten groups of experiments by randomly sampling five molecules from CROW dataset by ten times. For each group of experiment, we sample ten grammars for both hierarchical and non-hierarchical versions respectively and construct the geometry to cover the five molecules. The non-hierarchical grammar is sampled using the algorithm from Guo et al. (2022). The geometry construction is achieved following the BFS procedure described in Section 3.1 for both kinds of

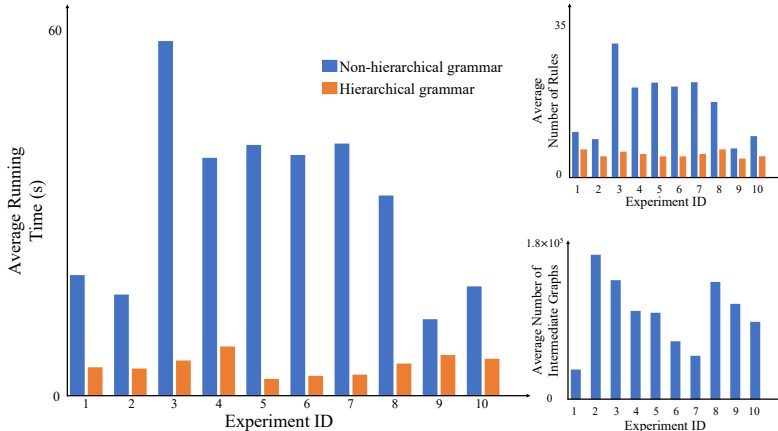

Figure 6: Analysis on construction cost reduction of hierarchical molecular grammar. For our hierarchical molecular grammar, the number of intermediate graphs is the same as the number of molecule samples, which is five in the experiments.

grammar. In the experiments, we find that for non-hierarchical grammar, even for five molecules, it occurs frequently that the size of the geometry grows extremely large but still cannot cover the five molecules. Therefore, we stop the construction when the size of the geometry reaches $2 \times 10^5$. Figure 6 illustrates (a) the average running time, (b) the average number of production rules, and (c) the average number of intermediate graphs of the final constructed geometry. It can be noted that our hierarchical molecular grammar greatly reduces the running time of the geometry construction. For non-hierarchical grammars, it is always intractable to construct the geometry since the number of intermediate graphs is enormous (some approaching $1.5 \times 10^5$). Our hierarchical molecular grammar provides a practical framework for grammar-induced geometry with theoretical foundations.

## D ANALYSIS ON COMPUTATIONAL COMPLEXITY

The computation of our approach consists of three parts: 1) the construction of grammar-induced geometry, 2) diffusion encoder, and 3) graph diffusion over the geometry. Following Blakely et al. (2021), suppose we have $K$ samples of molecules, each with $N$ nodes and $|\mathcal{E}|$ edges. To construct the grammar-induced geometry, we need to sample all the edges in order to contract molecules into junction trees, so the computational complexity is $\mathcal{O}(K|\mathcal{E}|)$. For diffusion encoder, the complexity is the same with general graph neural networks, which is $\mathcal{O}(KLd(Nd+|\mathcal{E}|))$, where $L$ is the number of layeres and $d$ is the feature dimension. As indicated in Chamberlain et al. (2021), the complexity of graph diffusion is $\mathcal{O}(|\mathcal{E}'|d)(E_b + E_f)$, where $\mathcal{E}'$ is the edge set of grammar-induced geometry, and $E_b$, $E_f$ are the numbers of function evaluations for forward and backward pass, respectively. Therefore, the overall computational cost of our method is $\mathcal{O}(K|\mathcal{E}| + KLd(Nd+|\mathcal{E}|) + |\mathcal{E}'|d(E_b + E_f)) = \mathcal{O}(KLd(Nd + |\mathcal{E}|) + |\mathcal{E}'|d(E_b + E_f))$. Compared with general graph neural networks which have a computational complexity of $\mathcal{O}(KLd(Nd + |\mathcal{E}|))$, our approach entails additional computations $\mathcal{O}(|\mathcal{E}'|d(E_b+E_f))$, where $|\mathcal{E}'|$ is the sum of the number of edges in the meta geometry $|\mathcal{E}_{meta}|$ and the number of samples $K$, since each molecule is connected to the meta geometry by an edge. In our experiments, we use a 4-degree meta grammar with depth 10, which gives us a meta geometry of 455 edges, i.e. $455 = |\mathcal{E}_{meta}| \ll K|\mathcal{E}|$. We also find in our experiments that $E_b + E_f \ll LNd$. As a result, the overall computational complexity of our method is the same as that of general graph neural networks in the big $\mathcal{O}$ sense. In practice, geometry can be constructed in parallel, which reduces the running time of our entire method.

## E DETAILS ON DATASETS

The Chemical Retrieval on the Web (CROW) polymer database[2] curates a database of thermo-physical data for over 250 different polymers, focusing primarily on the most commonly used plastics and resins in industry. CROW distinguishes from other digital polymer property databases due

---

[2]https://polymerdatabase.com/

to its heavy usage of real, experimental data individually sourced via extensive literature search. In the CROW polymer database, approximately $95\%$ of reported polymers offer some experimental data from literature, with the remaining $5\%$ derived purely from simulation. We only use those data with labels from real experiments in our evaluation. The CROW polymer database reports several properties including the glass transition temperature, Hildebrand solubility parameter, molar heat capacity, refractive index, and molar cohesive energy. Of these, we choose glass transition temperature as our benchmark performance comparison since it is one of the best documented polymer properties in the literature (Tao et al., 2021b). For each dataset except ClinTox, we randomly split the data into $4 : 1$ training and testing sets and create five such splits using $5$ random seeds. For ClinTox, we follow the settings in Zhou et al. (2022) to train our model and report the results of other baselines directly from the paper. Each of five splits is used to train and test a separate instance of each benchmark model. For the other three datasets, we refer the reader to the original papers: Permeability (Yuan et al., 2021), FreeSolv (Mobley & Guthrie, 2014), Lipophilicity (Wang et al., 2015), HOPV (Lopez et al., 2016), DILI (Ma et al., 2020), and PTC (Xu et al., 2018).

## F  DETAILS ON THE IMPLEMENTATION

**Baselines.** We compare our approach with various baselines: 1) Random Forest and FFN, two of the best-performing machine learning models for polymer informatics benchmarking (Tao et al., 2021b); 2) wD-MPNN, a state-of-the-art method specifically designed for polymer property prediction (Aldeghi & Coley, 2022); 3) ESAN, a general GNN architecture with enhanced expressive power (Bevilacqua et al., 2022); 4) HM-GNN, a motif-based GNN for molecular feature representation (Yu & Gao, 2022); 5) PN and Pre-trained GIN, two pre-trained GNNs with state-of-the-art performance on few-shot learning (Stanley et al., 2021) and transfer learning (Hu et al., 2020). As HM-GNN and Pre-trained GIN only provide code for graph classification, we modify the final layer of their networks and use $l_1$ loss for training. For the other methods, we follow the same implementation as their original papers. Since wD-MPNN cannot be depolyed on general molecules other than polymers, we report the results of D-MPNN (Yang et al., 2019) instead for two large datasets.

**Our System.** For our approach, we use 4-degree meta grammar, which contains eight rules. The meta geometry contains all the meta trees whose size is smaller than $11$, resulting in $149$ nodes and $455$ edges. For the molecular rule learning of $\theta$, we follow all the hyperparamters used in Guo et al. (2022). For the graph diffusion, the input feature of each meta tree node is the Weisfeiler Lehman graph hashing feature (Shervashidze et al., 2011). The encoder for meta tree nodes is an embedding layer that maps hashing features into a 300-dimension continuous vector. For molecular leaves, we use two different encoders: GIN from Xu et al. (2018) and MPNN from Yang et al. (2019), both of which output a feature vector of dimension 300. For the decoder, we use a one-layer fully connected network with size 300. For the graph diffusion process, we follow Chamberlain et al. (2021) and use Dormand–Prince adaptive step size scheme (DIORI5) with adjoint method. The diffusivity function $a(\cdot, \cdot; \alpha)$ is an attention function. We use Adam optimizer for the training of $\theta$ and $(\varphi, \psi, \alpha)$, with learning rate 0.01 and 0.001, respectively. We train $\theta$ for ten epochs. For each training epoch of $\theta$, we train $(\varphi, \psi, \alpha)$ for 50 epochs.

## G  RELATED WORKS

**Machine Learning for Molecular Property Prediction.** The use of machine learning methods to predict molecular properties has a long history. Many early methods use SMILES strings or handcrafted fingerprints as input and rely on traditional machine learning methods, such as random forest and Gaussian processes, which are still competitive today in many applications (Tao et al., 2021b; Joo et al., 2022). Recently, graph-based representations of molecules have gained increasing popularity with the development of GNNs (Feinberg et al., 2018; Xu et al., 2018; Wu et al., 2021; Bevilacqua et al., 2022; Yu & Gao, 2022; Aldeghi & Coley, 2022; Alon & Yahav, 2020). We refer the reader to Wieder et al. (2020) for a detailed review of GNN-based property predictors. Current state-of-the-art GNN-based methods provide more advanced representations of molecules based on graphs. Bevilacqua et al. (2022) represents each individual graph as a set of subgraphs, increasing the expressive power of GNNs. Aldeghi & Coley (2022) tailors molecular graphs by adding stochastic edges and designs GNN specifically for polymers based on Yang et al. (2019). Yu & Gao (2022) and Zhang et al. (2021) leverage motifs to model molecular relationships. All

these methods require large training datasets to achieve reasonable performance. It is common to use self-supervised learning (Rong et al., 2020; Wang et al., 2022) and transfer learning (Hu et al., 2020) to deal with sparse data, where neural networks are pre-trained on large datasets and then fine-tuned on the target small dataset. Most of these methods, however, focus solely on molecular classification, while our approach can also address regression problems with extremely sparse data, which are considerably more challenging.

**Molecular Grammars.** As an interpretable and compact design model, grammar has recently gained increasing attention in the field of molecule discovery (Dai et al., 2018; Kajino, 2019; Krenn et al., 2019; Xu et al., 2020; Nigam et al., 2021; Guo et al., 2021a; 2022). A molecular grammar can be considered as a generative model that uses production rules to generate molecules. The rules of a grammar can be either manually constructed (Dai et al., 2018; Krenn et al., 2019; Guo et al., 2021a) or automatically learned from molecular datasets (Kajino, 2019; Xu et al., 2020; Nigam et al., 2021; Guo et al., 2022). Recent works have demonstrated that a learnable hypergraph grammar can be effective in capturing hard chemical constraints, such as valency restrictions (Kajino, 2019; Guo et al., 2022). Guo et al. (2022) further proposes a learning pipeline for constructing a hypergraph grammar-based generative model that can incorporate domain-specific knowledge from very small datasets with dozens of samples. Despite the inherent advantages of molecular grammars, such as explicitness, explanatory power, and data efficiency, existing works mainly use them for molecular generation. Prediction and optimization of molecular properties can only be accomplished by using a separate model based on the grammar representation in a latent space, which is learned individually (Kajino, 2019; Xu et al., 2020). We integrate molecular grammar into property prediction tasks by constructing a geometry of molecular graphs based on the learnable grammar, which allows us to optimize both the generative model and the property predictor simultaneously. On extremely small datasets, our approach benefits especially from the data efficiency of grammar and achieves superior performance over existing property predictors.

**Hierarchical Molecular Generation.** The decomposition of molecular structures in our hierarchical molecular grammar is related to substructure-based methods in molecular generation (Jin et al., 2018; 2020; Maziarz et al., 2021). Jin et al. (2018) employs encoders and decoders to generate a junction tree-structured scaffold and then combine chemical substructures into a molecule. Jin et al. (2020) uses a hierarchical graph encoder-decoder to generate molecules in a coarse-to-fine manner, from atoms to connected motifs. Maziarz et al. (2021) integrates molecule fragments and atom-by-atom construction to generate new molecules. Our approach is fundamentally different from all these existing hierarchical representations in two respects: 1) *Without* the need for any training, our proposed meta grammar (the coarse level) can enumerate all possible junction tree structures by using a compact set of meta production rules, which can be *theoretically guaranteed*. Our method only requires learning at the fine level, which are molecular fragments determined by molecular rules, whereas existing methods require learning two models for both coarse and fine levels. 2) As opposed to existing methods that use latent spaces, our meta geometry induced by meta grammar explicitly models the similarity between molecules by using graph distance along the geometry. Due to the edit-completeness of meta grammar in Definition 1, the graph distance is an explainable measurement of minimal editing distance between molecular graphs, whereas the distance in latent spaces used in existing methods lacks explainability.

**Geometric Deep Learning** applies deep neural networks to non-Euclidean domains such as graphs and manifolds with a wide range of applications (Bronstein et al., 2017; Cao et al., 2020; Bronstein et al., 2021). Related to our method, a series of recent works use graph neural diffusion by treating GNNs as a discretisation of an underlying heat diffusion PDE (Chamberlain et al., 2021; Elhag et al., 2022; Bodnar et al., 2022). These methods work on large graph data where the graph connectivity of individual nodes is provided. Additionally, there are also existing works on inferring the underlying geometry of graph data. Ganea et al. (2018) and Cruceru et al. (2021) embed graphs into non-Euclidean manifolds such as hyperbolic and elliptical spaces and optimize their embeddings within these Riemannian spaces. Cosmo et al. (2020) learns a latent graph to model the underlying relationship between data samples and applies GNN to it. Different from these methods, our approach uses explicit and learnable intrinsic geometry based on graph grammar to model the relationship between molecular data. Since graph grammar is a generative model, our framework is capable of optimizing both molecular generation and property prediction simultaneously.

## H More Details of Hierarchical Molecular Grammar

**Discussion on Hierarchical Molecular Grammar.** The three additional attributes in Definition 1 can be used to deduce several desirable properties of a meta grammar. "Degree $k$" ensures the meta grammar is expressive and complete, covering all possible trees under a simple tree degree constraint. "Edit completeness" enables the explicit capture of transformations between two trees with edit distance one and therefore between two arbitrary trees with arbitrary distances. "Minimality" guarantees that the meta grammar is compact. Based on these three attributes, we can construct a generic meta grammar generating trees of non-terminal nodes by using only a small but expressive set of production rules. Figure 3(b) shows a 3-degree meta grammar. Each rule has one non-terminal node $\mathcal{R}^*$ on the $LHS$ and two $\mathcal{R}^*$s on the $RHS$. Depending on the number of anchor nodes on the $LHS$, these rules can transform tree nodes of different degrees by attaching a new node using different schemes indicated by the $RHS$. It is evident that we can generate any possible tree with a degree smaller than 4 by adopting a sequence of rules from this 3-degree meta grammar. In Proposition 2 "Completeness" states that any molecular graph can be derived from a hierarchical molecular grammar with a set of appropriate molecular rules. This can be demonstrated by the fact that we can generate an arbitrary molecule by first using the meta grammar to generate a tree of non-terminal nodes (which has the same tree structure as a junction tree decomposed from the molecule), and then using the molecular grammar to transform the tree into a molecular hypergraph by specifying a molecular fragment for each non-terminal node.

**Meta Rules Construction.** We refer to the production rule set of $k$-degree, edit-complete, and minimal meta grammar as a *$k$-degree meta rule set*. We visualize 4-degree meta rules in Figure 7a, where $\{p_1\}$, $\{p_1, p_2, p_3\}$, and $\{p_1, p_2, p_3, p_4, p_5\}$ correspond to meta rule sets of degree 1 to 3, respectively. 1-degree meta rule set $\mathcal{P}_{\overline{G}}^{(1)}$ is constructed as

$$
\mathcal{P}_{\overline{G}}^{(1)} = \mathcal{P}^{(1)} = \{p_0^{(1)}\},
$$
$$
p_0^{(1)} : LHS_0^{(1)} \rightarrow RHS_0^{(1)}, LHS_0^{(1)} := (\{\mathcal{X}\}, \varnothing), \ RHS_0^{(1)} := H(V_{R,0}^{(1)}, E_{R,0}^{(1)}),
$$
$$
V_{R,0}^{(1)} = \{\mathcal{R}_1^*, \mathcal{R}_2^*\}, E_{R,0}^{(1)} = \{(\mathcal{R}_1^*, \mathcal{R}_2^*)\}.
$$

When using the meta grammar for production, we treat $\mathcal{R}_1^*, \mathcal{R}_2^*$ as the same type of non-terminal node, i.e. $\mathcal{R}^* = \mathcal{R}_i^*, i = 1, 2$, despite the use of indices in the above formulation.

For $k$-degree meta rule set $\mathcal{P}_{\overline{G}}^{(k)}$ ($k > 1$), the construction is achieved by induction:

$$
\mathcal{P}_{\overline{G}}^{(k)} = \mathcal{P}_{\overline{G}}^{(k-1)} \cup \mathcal{P}^{(k)}, \ \mathcal{P}^{(k)} = \{p_0^{(k)}\} \cup \bigcup_{i=1}^{\lfloor \frac{k}{2} \rfloor} \{p_i^{(k)}\},
$$

$$
p_0^{(k)} : LHS_0^{(k)} \rightarrow RHS_0^{(k)}, \ LHS_0^{(k)} := H(V_{L,0}^{(k)}, E_{L,0}^{(k)}), \ RHS_0^{(k)} := H(V_{R,0}^{(k)}, E_{R,0}^{(k)}),
$$

$$
p_i^{(k)} : LHS_i^{(k)} \rightarrow RHS_i^{(k)}, \ LHS_i^{(k)} := H(V_{L,i}^{(k)}, E_{L,i}^{(k)}), \ RHS_i^{(k)} := H(V_{R,i}^{(k)}, E_{R,i}^{(k)}), i = 1, ..., \lfloor \frac{k}{2} \rfloor,
$$

$$
V_{L,0}^{(k)} = \{\mathcal{R}^*\} \cup \bigcup_{j=1}^{k-1} \{V_{anc,j}\}, \ E_{L,0}^{(k)} = \bigcup_{j=1}^{k-1} \{(\mathcal{R}^*, V_{anc,j})\},
$$

$$
V_{R,0}^{(k)} = \{\mathcal{R}_1^*, \mathcal{R}_2^*\} \cup \bigcup_{j=1}^{k-1} \{V_{anc,j}\}, \ E_{R,0}^{(k)} = \{(\mathcal{R}_1^*, \mathcal{R}_2^*)\} \cup \bigcup_{j=1}^{k-1} \{(V_{anc,j}, \mathcal{R}_2^*)\},
$$

$$
V_{L,i}^{(k)} = \{\mathcal{R}^*\} \cup \bigcup_{j=1}^{k} \{V_{anc,j}\}, \ E_{L,i}^{(k)} = \bigcup_{j=1}^{k} \{(\mathcal{R}^*, V_{anc,j})\},
$$

$$
V_{R,i}^{(k)} = \{\mathcal{R}_1^*, \mathcal{R}_2^*\} \cup \bigcup_{j=1}^{k} \{V_{anc,j}\}, \ E_{R,i}^{(k)} = \{(\mathcal{R}_1^*, \mathcal{R}_2^*)\} \cup \bigcup_{j=1}^{i} \{(V_{anc,j}, \mathcal{R}_1^*)\} \cup \bigcup_{j=i+1}^{k} \{(V_{anc,j}, \mathcal{R}_2^*)\}.
$$

Specifically, the $k$-degree meta rule set $\mathcal{P}_{\overline{G}}^{(k)}$ contains all the rules from $(k-1)$-degree meta rule set $\mathcal{P}_{\overline{G}}^{(k-1)}$ as well as other newly introduced rules $\mathcal{P}^{(k)}$. Each rule contains one non-terminal node $\mathcal{R}^*$

on the $LHS$ and two non-terminal nodes $\mathcal{R}_1^*$ and $\mathcal{R}_2^*$ on the $RHS$. The $LHS$ of $p_0^{(k)}$ has a degree of $k-1$ as it contains $k-1$ anchor nodes $V_{anc}$. The $RHS$ of $p_0^{(k)}$ attaches one non-terminal node $\mathcal{R}_2^*$ to the other non-terminal node $\mathcal{R}_1^*$ which connects all the anchor nodes. Therefore, the $RHS$ has a degree of $k$ which is larger than the degree of the $LHS$ by one. In all the other rules, the $LHS$ has $k$ anchor nodes and is of degree $k$ while the $RHS$ keeps a maximal degree of $k$. The $k$ anchor nodes on the $RHS$ are combinatorially distributed to $\mathcal{R}_1^*$ and $\mathcal{R}_2^*$, i.e. if $\mathcal{R}_1^*$ is attached to $i$ anchor nodes, $\mathcal{R}_2^*$ is attached to the rest of $k-i$ anchor nodes. Since we consider $\mathcal{R}_1^*$ and $\mathcal{R}_2^*$ to be the same during grammar production, the range of $i$ is $\{1, ..., \lfloor \frac{k}{2} \rfloor\}$. In total, there are $1+\lfloor \frac{k}{2} \rfloor$ rules in $\mathcal{P}^{(k)}$.

*Proof of Edit Completeness.* According to the definition in Zhang (1996); Paaßen (2018), tree edit distance $TED(T, T')$ between two trees $T$ and $T'$ is the minimum number of operations required to transform one tree into the other. There are three types of available operations: inserting, deleting, and relabeling. Consider two arbitrary trees $T$ and $T'$ that satisfy the definition of edit completeness (being derived from the meta grammar, $TED(T, T') = 1$, and $|T| < |T'|$). Since all the trees generated from the meta grammar have homogeneous tree node labels (i.e. $\mathcal{R}^*$), there is no relabeling operation in $TED$ calculation. Furthermore, the tree size relation between $T$ and $T'$ admits only one type of tree edit operation: a one-step inserting. Figure 7b illustrates the one-step inserting operation. It operates on a certain node $v$ in $T$ whose subtree is $T_{sub}(v)$. A new node $v'$, which takes a subset of $T_{sub}(v)$ as the children, is added as a new child of node $v$. The rest of $T_{sub}(v)$ stays as the children of node $v$. Suppose the degree of node $v$ in $T$ is $m$ and the degree related to subtree $T_{sub}(v)$ is $n$. Without loss of generality, we suppose $0 \le n \le m/2$. We can treat $v$ in $T$ as the $\mathcal{R}^*$ on the $LHS$, $v'$ and $v$ in $T'$ as $\mathcal{R}_1^*$ and $\mathcal{R}_2^*$ on the $RHS$, respectively. Then the one-step inserting operation can be achieved using the rule $p_n^{(m+1)} \in \mathcal{P}^{(m+1)}$ from $(m+1)$-degree meta rule set $\mathcal{P}_{\overline{G}}^{(m+1)}$ as constructed above. Since $\Delta(T') \ge m+1$, the meta grammar $\mathcal{P}_{\overline{G}}^{(k)}$ that generates $T'$ has $k \ge m+1$. So $p_n^{(m+1)} \in \mathcal{P}^{(m+1)} \subseteq \mathcal{P}_{\overline{G}}^{(k)}$; thus finish the proof.

*Proof of Degree $k$.* For an arbitrary tree $T$ with maximal degree $k$, we prove that meta grammar $\mathcal{P}_{\overline{G}}^{(k)}$ can generate $T$ by finding a reverse path of production rule from the tree $T$ to the initial node $\mathcal{X}$. We first find all the nodes $\{v_i\}_{i=1}^n$ in $T$ that have a degree of $k$. For each node, by edit completeness, we can use the inverse operation of inserting and make the node to have a degree of $k-1$. It can be shown from the proof for edit completeness that each inverse operation of inserting corresponds to an inverse deployment of a production rule. By using $n$ reverse production steps on all the $k$-degree nodes, we can obtain a $T'$ with $\Delta(T') = k-1$. We can continue the process until the tree is 0-degree, i.e. the $\mathcal{X}$; thus finish the proof.

*Proof of Minimality.* From edit completeness, we know each production rule in the meta grammar corresponds to at least one case of one-step inserting operation. As constrainted by degree $k$, arbitratry trees that are of degree smaller than $k$ need to be covered by the meta grammar. We thus need to handle all the possible cases of one-step inserting operation when finding the reverse production path from the tree to $\mathcal{X}$. Note that one-step inserting opeartion involves the division of a subtree, which has combinatorial ways. Then every combinatorial distribution of anchor nodes corresponds to one case of subtree division. This proves each production rule in the meta rule set is necessary. Since there are no duplicate rules, it can be concluded that the meta grammar constructed above is minimal.

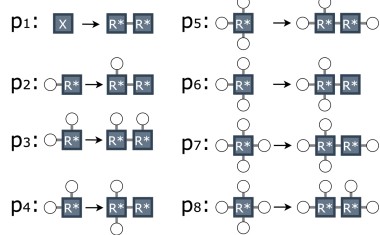

(a) 4-degree meta rules.

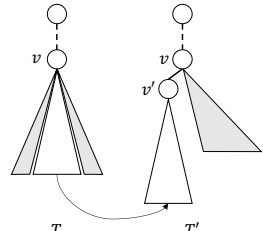

(b) One-step inserting operation.

Figure 7: Illustration of inserting operation and 4-degree meta grammar.

**4-degree Meta Rules.** Figure 7a shows the 4-degree meta grammar we use in our approach. Since the edit completeness is the foundation to prove all the other attributes of the meta grammar, we use a tree edit distance toolkit[3] and verify its correctness for the 4-degree meta grammar.

---

[3] https://pythonhosted.org/zss/

## I  MOLECULAR RULES CONSTRUCTION

---

**Algorithm 1:** Molecular Rules Construction.

---

**Input** : molecular hypergraph $H = (V, E_H)$, probability function $\phi(\cdot; \theta)$

**Output:** molecular rule set $\mathcal{P}_{mol} = \{p_i\}_{i=1}^M$, junction tree $T = (V_T, E_T)$ where each $v_T \in V_T$ is a subset of $V$

1  Initialize $\mathcal{P}_{mol} = \varnothing$, $V_T = \varnothing$, $E_T = \varnothing$;

2  Create a set for unvisited nodes $V_{unv} \leftarrow V$, a set for unvisited hyperedges $E_{unv} \leftarrow E_H$;

3  **while** $E_{unv} \neq \varnothing$ **do**

4     $\mathbf{e} = [e_1, ..., e_K]$, $e_i \in E_{unv}$, $i = 1, ..., K = |E_{unv}|$;

5     $\mathbf{X} \sim \text{Bernoulli}(\phi(\mathbf{e}; \theta))$;

    /* Construct a rule for each connected component         */

6     **for** each $H_{sub} = (V_{sub}, E_{sub})$ in `ConnectedComponents`($\mathbf{X}$, $H$) **do**

        /* Separate the connected component             */

7         $\hat{V}_{sub} \leftarrow V_{sub}$, $\hat{E}_{sub} \leftarrow E_{sub}$, $V_{anc} \leftarrow \varnothing$, $E_{anc} \leftarrow \varnothing$;

8         **for** each $v \in V_{sub}$ **do**

9             $V_n = $ `GetNeighbors`($v$, $H$);

10            **if** $V_n \cap V_{sub} \neq \varnothing$ **then**

11                **if** $v \notin V_{unv}$ or `InRing`($v$, $H$) **then**

12                    $\hat{V}_{sub} \leftarrow \hat{V}_{sub} \setminus \{v\}$;

13                    $\hat{E}_{sub} \leftarrow \hat{E}_{sub} \setminus \{(s, v) | (s, v) \in E_{sub}\}$;

14                    $V_{anc} \leftarrow V_{anc} \cup \{v_{anc}^{(i)}, i = |V_{anc}| + 1\}$;

15                    $E_{anc} \leftarrow E_{anc} \cup \{(s, v_{anc}^{(i)}) | (s, v) \in E_{sub}, i = |V_{anc}| + 1\}$;

16                **else**

17                    $V_{anc} \leftarrow V_{anc} \cup \{v_{anc}^{(i)}, i = |V_{anc}| + k\}_{k=1}^{|V_n \cap V_{sub}|}$;

18                    $E_{anc} \leftarrow E_{anc} \cup \{(s, v_{anc}^{(i)}) | (s, v) \in E_{sub}, i = |V_{anc}| + k\}_{k=1}^{|V_n \cap V_{sub}|}$;

        /* Construct a production rule             */

19         $p = $ `RuleConstruction`($\hat{V}_{sub}$, $\hat{E}_{sub}$, $V_{anc}$, $E_{anc}$);

20         $\mathcal{P}_{mol} \leftarrow \mathcal{P}_{mol} \cup \{p\}$;

        /* Construct junction tree               */

21         $v_T = (V_{sub})$;

22         **for** each $\{v_t \in V_T | v_t \cap v_T \neq \varnothing\}$ **do**

23             $E_T \leftarrow E_T \cup \{(v_t, v_T)\}$;

24         $V_T \leftarrow V_T \cup \{v_T\}$;

        /* update visited status of nodes and hyperedges     */

25         $V_{unv} \leftarrow V_{unv} \cup V_{sub}$, $E_{unv} \leftarrow E_{unv} \cup E_{sub}$;

26  **return** $\mathcal{P}_{mol}$, $T = (V_T, E_T)$;

---

To obtain the input for the molecular rule construction, we first convert the molecule into a molecular hypergraph $H = (V, E_H)$. A node $v \in V$ represents an atom of the molecule. A hyperedge $e \in E_H$ corresponds to either a bond that joins only two nodes or a ring (including aromatic ones) that joins all nodes in the ring. An illustration is provided in Figure 2 of Guo et al. (2022). The probability function $\phi(\cdot; \theta)$ is defined on each hyperedge: $\phi(e; \theta) = \sigma(-\mathcal{F}_\theta(f(e)))$, where $\sigma(\cdot)$ is the sigmoid function, $\mathcal{F}_\theta(\cdot)$ is a two-layer fully connected network whose final output dimension is 1, and $f(\cdot)$ is a feature extractor using a pre-trained GNN (Hu et al., 2020).

Algorithm 1 illustrates the detailed process of constructing molecular rules for a single molecule. Note that in our approach, the molecular rule construction is performed simultaneously for all input molecules. For each molecule, we first perform an i.i.d. sampling on all the hyperedges following a Bernoulli distribution which takes the value 1 with probability indicated by $\phi(e; \theta)$. We then obtain a binary vector $\mathbf{X}$ that indicates whether each hyperedge is sampled (line 4-5). Next, all connected components are extracted with respect to the sampled hyperedges (line 6). A production rule is constructed for each connected component $H_{sub} = (V_{sub}, E_{sub})$ (line 7-19). Specifically, a production rule needs two components: a hypergraph $\hat{H}_{sub} = (\hat{V}_{sub}, \hat{E}_{sub})$ for *RHS* and anchor

nodes $V_{anc}$ indicating the correspondence between the $LHS$ and the $RHS$. $\hat{H}_{sub}$ is obtained by removing visited nodes and in-ring nodes from $H_{sub}$. $V_{anc}$ contains nodes from $V$ that are connected to nodes from $H_{sub}$ but do not appear in $H_{sub}$ themselves. $E_{anc}$ are the edges connecting anchor nodes to $\hat{H}_{sub}$ following the same connectivity in the original graph $H$. Following Guo et al. (2022), the function `RuleConstruction`$(\hat{V}_{sub}, \hat{E}_{sub}, V_{anc}, E_{anc})$ returns a production rule $p : LHS \rightarrow RHS$ constructed as

$$
\begin{aligned}
LHS &:= H(V_L, E_L), V_L = \{\mathcal{R}^*\} \cup V_{anc} \,, \; E_L = \{(\mathcal{R}^*, v) | v \in V_{anc}\} \,, \\
RHS &:= H(V_R, E_R), V_R = \hat{V}_{sub} \cup V_{anc} \,, \; E_R = \hat{E}_{sub} \cup E_{anc}.
\end{aligned}
\tag{3}
$$

In contrast to Guo et al. (2022), our molecular rule construction does not replace the connected component with the non-terminal node $\mathcal{R}^*$ at every iteration since according to our definition, the molecular rule does not contain any non-terminal nodes on the $RHS$.

The junction tree of the molecule is constructed along with the construction of molecular rules (line 21-24). We simply treat each connected component $V_{sub}$ sampled at each iteration as a node $v_T \in V_T$ in the junction tree $T = (V_T, E_T)$. Two nodes are connected by an edge $e_T \in E_T$ if their corresponding connected components share hypergraph nodes from $H$. Since each hyperedge in $H$ is only visited once, the constructed junction tree then contains all the hyperedges and nodes of $H$ without redundancy (Kajino, 2019).

## J GRAMMAR-INDUCED GEOMETRY CONSTRUCTION

---

**Algorithm 2:** Grammar-induced Geometry Construction.

---

**Input** : meta production rules $\mathcal{P}_{\overline{G}} = \{p_i\}_{i=1}^N$, maximum BFS depth $D$, a set of molecular hypergraphs $\{H_i\}_{i=1}^M$ and their corresponding junction trees $\mathcal{J} = \{T_i\}_{i=1}^M$
**Output:** geometry in the form of a graph $\mathcal{G} = (\mathcal{V}, \mathcal{E})$ where each $v = H_v = (V_v, E_v) \in \mathcal{V}$ represents a meta tree or a molecular hypergraph and $\mathcal{E}$ is the edge set of the geometry

    /* Add root of the geometry                                */
1 Initialize $v_{root} = H_{root} = (\mathcal{X}, \varnothing), \mathcal{V} = \{v_{root}\}, \mathcal{E} = \varnothing$;
2 Create a queue data structure $Q$;
3 $Q$.equeue$(v_{root})$;
    /* Breadth-first search for meta geometry, pre-computed offline     */
4 **while** $Q$ is not empty and $|\mathcal{V}| \leq D$ **do**
5      $v = Q$.dequeue$()$;
        /* Expand the meta geometry                                */
6      **for** each $p_i \in \mathcal{P}_{\overline{G}}$ **do**
7          **if** $p_i$ is applicable to $H_v$ **then**
8              $H_v \overset{p_i}{\Rightarrow} H_{new}$;
9              $v_{new} = H_{new}$;
10              **if** $v_{new} \notin \mathcal{V}$ **then**
11                  $\mathcal{V} \leftarrow \mathcal{V} \cup \{v_{new}\}, \mathcal{E} \leftarrow \mathcal{E} \cup \{(v, v_{new})\}$;
12                  $Q$.equeue$(v_{new})$;
13              **else**
14                  $\hat{v} = $ `GetIsomorphicGraph`$(H_{new}, \mathcal{V}), \mathcal{E} \leftarrow \mathcal{E} \cup \{(v, \hat{v})\}$;

    /* Connect molecular leaves during run-time                      */
15 **for** each $T_i \in \mathcal{J}$ **do**
16      **for** each $v \in \mathcal{V}$ **do**
17          **if** `IsIsomorphic`$(H_v, T_i)$ **then**
18              $v_i = H_i$;
19              $\mathcal{V} \leftarrow \mathcal{V} \cup \{v_i\}, \mathcal{E} \leftarrow \mathcal{E} \cup \{(v, v_i)\}$;
20 **return** $\mathcal{G} = (\mathcal{V}, \mathcal{E})$;

---

Algorithm 2 illustrates the detailed algorithm to construct the grammar-induced geometry. It contains two parts: the construction of the meta geometry (line 4-14) and the construction of the molecular leaves (line 15-19). The meta geometry construction follows the standard breadth-first search

(BFS) starting from the root $H_{root} = (\mathcal{X}, \varnothing)$ (line 1). Every time when we visit a node $v \in \mathcal{V}$ in $\mathcal{G}$, we find all the rules that are applicable to $H_v$ from the meta rule set (line 6-7). Each applicable rule is applied to $H_v$ in order to create a new meta tree $H_{new}$ (line 8). Depending on whether there is a node in $\mathcal{V}$ that represents a isomorphic tree to $H_{new}$ in the current geometry, we either create a new node $v_{new}$ (line 9) or find the existing matched node $\hat{v}$ (line 14). We then add an edge between $v$ and $v_{new}$, or between $v$ and $\hat{v}$ (line 11 and 14). The function `GetIsomorphicGraph`$(H_{new}, \mathcal{V})$ enumerates every node in $\mathcal{V}$, checks if the tree represented by the node is isomorphic to $H_{new}$, and returns the matched node. We use the algorithm from Cordella et al. (2001) which is implemented in `networkx`[4] for graph isomorphism test. To increase the speed of `GetIsomorphicGraph`$(H_{new}, \mathcal{V})$, we use Weisfeiler Lehman graph hash (Shervashidze et al., 2011) and only perform isomorphism test for graph pairs that share the same hashing code.

The molecular leaves are added during run-time (line 15-19). For each input molecular hypergraph, we check if there is a meta tree node that is isomorphic to its junction tree. If so, we add an edge connecting the matched meta tree node to the molecular hypergraph. The function `IsIsomorphic`$(H_v, T_i)$ is implemented using the same package for `GetIsomorphicGraph`$(H_{new}, \mathcal{V})$ and returns true if $H_v$ and $T_i$ are isomorphic to each other.

## K  OPTIMIZATION

Figure 8 shows a graphical model of dependency in our optimization problem. The two sets of parameters $\theta$ and $(\varphi, \psi, \alpha)$ are independent and thus can be iteratively optimized using block coordinate descent. Since the objective in Equation 2 is differentiable with respect to $(\varphi, \psi, \alpha)$, we can use gradient descent for the optimization iterations related to the graph neural diffusion. The geometry learning, however, is non-differentiable due to the fact that we perform sampling to construct molecular rules and construct the geometry. Hence, we rewrite the objective for optimizing $\theta$ in an expectation form and apply REINFORCE (Williams, 1992) to obtain a stochastic gradient, as done in Guo et al. (2022):

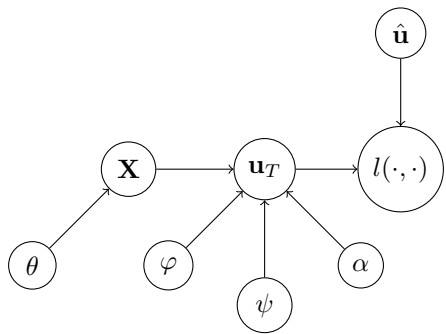

Figure 8: A graphical model of dependency.

$$\min_{\theta} l(\mathbf{u}_T, \hat{\mathbf{u}}) = \min_{\theta} \mathbb{E}_{\mathbf{X}}\Big[l(\mathbf{u}_T, \hat{\mathbf{u}})\Big],$$

$$\nabla_{\theta}\mathbb{E}_{\mathbf{X}}\Big[l(\mathbf{u}_T, \hat{\mathbf{u}})\Big] = \int_{\mathbf{X}} l(\mathbf{u}_T, \hat{\mathbf{u}})\nabla_{\theta}p(\mathbf{X})$$

$$= \mathbb{E}_{\mathbf{X}}\Big[l(\mathbf{u}_T, \hat{\mathbf{u}})\nabla_{\theta}\log(p(\mathbf{X}))\Big] \approx \frac{1}{N}\sum_{n=1}^{N} l(\mathbf{u}_T^{(n)}, \hat{\mathbf{u}})\nabla_{\theta}\log(p(\mathbf{X}^{(n)})),$$

where $\mathbf{X}$ is a concatenation of binary vectors indicating how hyperedges are sampled in molecular rule construction in Appendix I.

## L  EXAMPLES OF RETRO-SYNTHESIS PATHS

Figure 9 shows the retro-synthesis paths of three molecules generated using our pipeline. Since Retro[*] score is used as one of the grammar metrics, our approach is capable of retro-synthesis planning for all the generated molecules, providing a complete pipeline for novel molecule discovery.

---

[4]`https://networkx.org/documentation/stable/reference/algorithms/isomorphism.html`

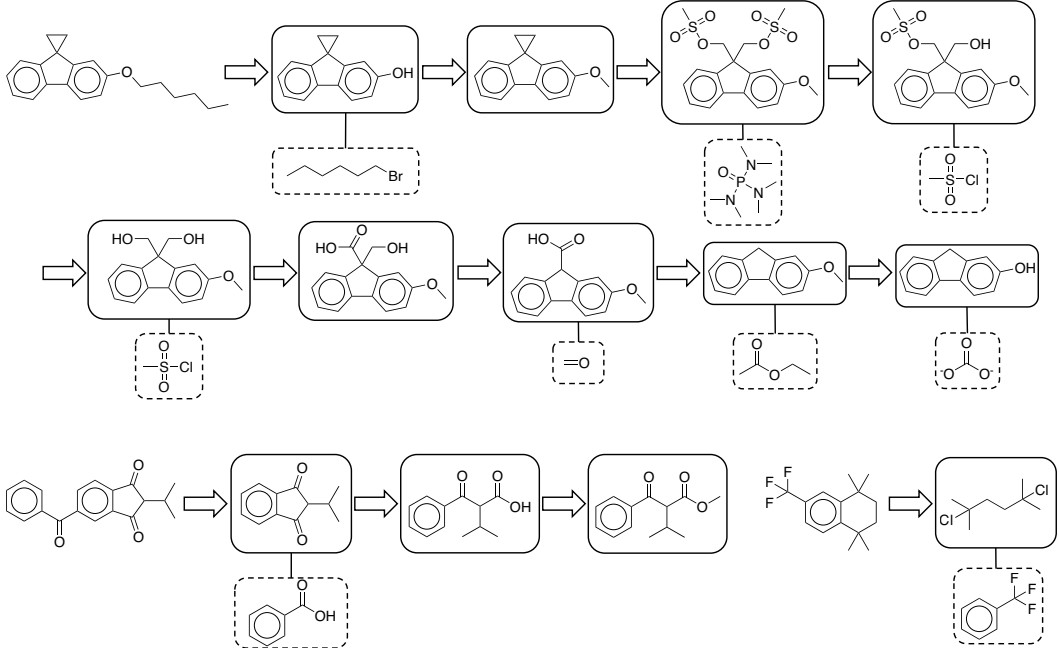

Figure 9: Examples of retro-synthesis paths.

