# OpenReview forum: "Grammar-Induced Geometry for Data-Efficient Molecular Property Prediction"
_ICLR.cc/2023/Conference — Submitted to ICLR 2023_

### Official Review · Reviewer_aHZ7 · 2022-10-14

**Confidence:** 4
**Correctness:** 3
**Technical Novelty And Significance:** 2
**Empirical Novelty And Significance:** 2
**Recommendation:** 6

**Clarity, Quality, Novelty And Reproducibility:**

Quality:
The work is of good quality in terms of the ideas. The descriptions of the model are also clear and thorough. However, the depth of the empirical evaluation and support of the claims is not as strong as it could be.

Clarity:
Overall, the paper is clear and easy to read.
In Figure 4(b) it would be helpful to use number of datapoints to label the axes, rather than percentages, in line with the few-shot literature.

Originality:
The work brings some original ideas, notably using hierarchical molecular grammars in molecular property prediction, and the use of graph neural diffusion for this purpose. The hierarchical grammar is an additional useful contribution above previous work with no hierarchy, although similar ideas have been seen in molecular graph generation eg. JT-VAE. This work clearly follows the lead of other recent works on molecular grammars and induced geometries and has strong similarity in many of the foundational ideas, so in that sense is extending that area of literature. The main novelty is the application of the ideas in molecular property prediction.

**Strength And Weaknesses:**

Strengths:
* This is a nice extension of molecular grammars to include a more computationally tractable method by inclusion of a hierarchical component.
* The use of graph neural diffusion to make use of the grammar-induced geometry for molecular property prediction is a good idea -- one would indeed hope that a representation using explicit molecular substructures would ultimately be the best for molecular comparison, and it is helpful that the authors explore this.
* The description of the model is clear.
* The level of detail given in the appendix regarding the molecular rules and geometry construction is helpful.

Weaknesses:
* The lack of reference to related work -- the section in the appendix only addresses very general molecular property prediction and not molecular grammars, which in the context of a number of other works approaching similar ideas seems lacking. Concretely, such a section would be highly valuable. In addition, while Guo et al. 2022 is referenced, it does not seem that the strong similarities are acknowledged. Is it possible to compare the performance of this model to one using a non-hierarchical grammar (i.e. does this particular hierarchical grammar improve the molecular property prediction)? In addition, reference is made to the superior computational performance; is it possible to quantify this further? Such comparisons would be very useful. Furthermore, there have been previous works taking a hierarchical approach to the decomposition of molecular structure, such as JT-VAE, and hierVAE. While these works are in molecular generation, the underlying principles bear strong similarity. Extension of the related work section (and possible inclusion in the main body of the paper) would be helpful.
* The authors strictly address only regression. While this is certainly a more challenging task in general, it does a) make direct comparison to other methods more challenging b) limit the range of datasets on which the method can be evaluated. The small molecule datasets in use appear to be the more challenging regression ones in Molecule Net (i.e. excluding quantum mechanics). As the authors note in the appendix, most related work in molecular property prediction is in classification, so it would be helpful to be able to draw some direct comparison to the state-of-the-art methods there.
* While the authors note that molecular grammars should improve the ability for a model to generalise outside of the domain, and give this as a reason for improved performance of their method over pre-trained GIN and PN, as these were trained on lower average molecular weight datasets than Permeability, they do not justify this with their results. In each of the CROW and Permeability cases their method was trained on the dataset at hand, and so does not have to demonstrate cross-domain adaptation. While GIN and PN are fine-tuned on the datasets, they come with strong intrinsic bias to their original pretraining datasets, and were pretrained for classification in addition. As both were pretrained on a set of small drug-molecule protein-binding data, it is actually remarkable they perform so well on a polymer dataset, where the relevant substructures are likely quite different. A concrete improvement would be demonstration of cross-domain transfer of the model.
* Overall, the empirical evaluation and discussion felt rushed, the paper would benefit from more expansion around the results. Given the explicit reference to four key questions in the evaluation section, it would be helpful to have these more clearly addressed in the main body of the text, and perhaps slightly less time in description of the model.
* Section 4.4 "Combination with generative models" is not a strong comparison to the other work in molecular generation, it is not clear how this particular method really compares in performance to, for instance, Guo et al 2022. It is also generally unclear whether the method has made improvements, or this is simply demonstrating that this kind of generation can be done, which was already answered by previous works.
* Figure 4(b) addresses the effect of changing dataset training size on model performance.
  * This kind of plot has appeared with other datasets such as FS-Mol. Given that this exploration is very useful, it would be helpful if the authors included analysis of performance on other datasets.
  * A straight line represents the performance of a Pretrained-GIN, which is therefore presumably not finetuned on training set data. This is an unfair comparison. The Pretrained-GIN should be finetuned on the same data that the author's model has access to, and those results reported, in line with the typical few-shot literature eg. Hu et al. 2020, Stanley et al. 2021.
  * It would be helpful to see the performance of a few more of the benchmarked models as the training set size is changed.

**Summary Of The Paper:**

This paper addresses the problem of molecular property prediction, particularly focusing on the limited availability of labelled data in many domains. It notes the limitations of existing methods targeted towards this scenario, and presents a molecular hypergraph grammar to describe molecules, and this grammar is used to induce a geometry over molecular space to enable effective molecular similarity measurement. To combat the computational challenge of enumerating a full molecular grammar-induced geometry the authors propose a hierarchical grammar consisting of a junction tree meta-geometry and a lower assignment of tree nodes to true molecular graphs. The property prediction then uses a graph neural diffusion model operating on all nodes in the grammar-induced geometry. The performance of this method is evaluated on two small datasets of polymers, and two slightly larger small molecule property prediction datasets.

**Summary Of The Review:**

The paper makes a good contribution with a novel method for molecular property prediction that uses a hierarchical grammar to induce a geometry, then using this to make property predictions. The idea of using such a grammar in a property prediction setting is novel, and the additional hierarchical component is somewhat novel but bears some similarities to previous work on molecular substructures such as JT-VAE, hierVAE, and MoLER. In addition, the molecular grammar itself is very similar to the work of Guo et al. 2022. Therefore, much of the additional contribution comes from the application of these ideas in property prediction rather than molecule generation. This is a nice potential use of such a framework, and indeed could result in better cross-domain adaptation, but it was not clear the authors fully explored or supported claims made in the paper around this. In addition, the limitation of property prediction tasks to regression feels somewhat odd -- while regression is indeed more challenging, not addressing the many unsolved classification problems available in datasets such as MoleculeNet and FS-Mol prevents a strong comparison between this work and previous works being made.

Overall, the ideas in this paper, while clearly strongly inspired by other works, are good. However, the paper in its current form is marginal because it would benefit from more time spent in experimentation and analysis to more strongly support the claims made. While the ideas are good it is not clear that the method is generally performing well, and whether it is more broadly applicable. If the method is truly very effective, and this can be demonstrated to be so on a wider range of commonly benchmarked tasks, then this would be distinctly impactful to the field. However, as it stands such conclusions are difficult to reach on the basis of the empirical exploration presented here.

====update in response to authors comments====
Following the response and revisions of the authors, in particular the inclusion of additional evaluation, a discussion of the computational complexity, and more clarification of the baselines, this review is being updated to reflect this.

---

> ### Author Response · Authors · 2022-11-10
> **Response to Reviewer aHZ7**
>
> **(Q1) Missing References.**
>
> Thank you for the suggestions! In Appendix G of the revised draft, we have added a new subsection that discusses molecular grammars in detail. We highlight the differences between our approach and existing grammar-based approaches by focusing on the difference in tasks: our method aims at property prediction whereas existing approaches are only designed for molecule generation. In Appendix G, we also provide a detailed comparison of our method with existing hierarchical approaches. See also **General Response** for more details.
>
> **(Q2) Comparison with non-hierarchical grammar.**
>
> As we mentioned in the ending paragraph of Section 3.1, it is computationally intractable to construct a grammar-induced geometry for non-hierarchical grammars. In Appendix C, we provide a quantitative comparison of the computational cost of the grammar from [Guo et al. 2022] and the hierarchical grammar we propose. Based on the analysis, we can see our hierarchical grammar greatly reduced the computational cost to the point where the induced geometry can be constructed online. For non-hierarchical grammars in [Guo et al. 2022], it is usually impossible to construct the geometry even for 5 data points, not to mention a larger training dataset. See also **General Response** for more details.
>
> **(Q3) More experiments on classification.**
>
> We have added three classification datasets (DILI, PTC, and ClinTox) to the experiments in Appendix A. Our approach still outperforms other baseline methods on all three classification tasks, which demonstrates the effectiveness of our method.
>
> **(Q4) Cross-domain transfer.**
>
> Our intention was to develop a model that is able to learn domain-specific features based on very small datasets, in line with the work of [Guo et al. 2022]. We would like to stress that **the generalizability of our approach comes from the fact that it can be trained for each individual dataset, however small, from scratch** (which we did for all the experiments), thus avoiding the domain gap.
> The pre-trained GIN and PN are used as baselines, **not** to compare their domain adaptation capability, but rather because the pretrain-finetuning strategy is commonly used for deep networks to achieve reasonable performance on small datasets. Therefore, they are strong baselines in the experiments.
>
> Though it is not our intention to show the cross-domain adaptation of our model, it is possible to transfer grammar-based knowledge between datasets by transfer learning the sampling function used to learn the grammar. This is beyond the scope of the current work but a very interesting direction for future pursuit.
>
>
> **(Q5) More evaluation and more discussion on results.**
>
> Thank you for the suggestions. We have added more discussions on the results in both the main text and Appendix A, addressing the four key questions. For other questions on the empirical evaluation, please see the **General Response**.
>
> **(Q6) "Combination with generative models".**
>
> The experiments in Section 4.4 are intended to address the last question we raised at the beginning of Section 4. We did not intend to compare with other molecular generative models, but rather to show that molecular generation (using the learned grammar) can be treated as a by-product of our framework and that joint optimization can be accomplished by simply adding grammar metrics to the objective and optimizing.
>
> **(Q7) Clarification of Figure 4(b).**
>
> We clarify that the Pre-trained GIN model in Figure 4(b) **is finetuned on the whole training data**. We have added a sentence to clarify this in the caption of Figure 4(b). Further analysis of a few other benchmarking models is provided in Appendix B. Across different training dataset sizes, our proposed method consistently outperforms other baselines. A smaller training dataset leads to a larger performance improvement, demonstrating the data efficiency of our model.

---

> > ### Comment · Reviewer_aHZ7 · 2022-11-27
> > **Response to authors**
> >
> > Thank you for your response to the comments.
> >
> > To address point (4) Cross-domain transfer: it makes sense that the approach is able to learn quickly the relevant features of small datasets from scratch. However, the wording around this is slightly misleading, as domain transfer does not typically refer to the ability of a model to *adapt* quickly to a new dataset, rather that the model should work on a new dataset without further finetuning.
> >
> > The use of the PN, which was pretrained on a large drug dataset, as a baseline in very unrelated tasks remains somewhat unrealistic as a baseline. A better baseline would be to pretrain such a model on a dataset of similar molecules to the tasks at hand. While I understand that the advantage of the proposed approach in this paper is that it does not need such pretraining, it is misleading to claim that such pretraining would not work on the basis of a model trained in a different domain -- these models would not be expected to work and do not make a suitable baseline.
> >
> > However, I recognise that in some domains such large datasets are not available to perform a comparison against a large pretrained model, and so there is a limitation here. In that case, it may be best to highlight directly that this is the case for clarity for future readers.
> >
> > On the basis of the discussion here, and the changes made in the paper (particularly around the analysis of computational complexity) I wish to revise my review to 6, marginally above the acceptance threshold.

---

### Official Review · Reviewer_EFAZ · 2022-10-17

**Confidence:** 2
**Correctness:** 3
**Technical Novelty And Significance:** 3
**Empirical Novelty And Significance:** 3
**Recommendation:** 6

**Clarity, Quality, Novelty And Reproducibility:**

The work and the proposed method is sufficiently described in the manuscript, and clear effort was spent in making the work understandable.

In terms of novelty, I believe this is the first work that adapts grammar-induced geometries for the task of molecular prediction, albeit these approaches had already been common in approaches such as JT-VAE for molecular generation.

In terms of reproducibility, while there is enough level of detail in both the main manuscript and appendix, the authors do not provide any code for either the proposed approach or to reproduce the results in the study.

**Strength And Weaknesses:**

Strengths:
* Natural extension of the molecular grammar work proposed by Guo et al. (2022) for the supervised learning case
* The methodology is well motivated and described
* Good level of detail provided in the appendix


Weaknesses:
* Rushed experiment section. Specifically the authors limit themselves to a couple of relatively small datasets when they could test on more widely used benchmarks in molecular ML to demonstrate the superiority of their approach (see summary of the review).
* Computational complexity not clear.
* No code provided to reproduce results / test the existing approach

**Summary Of The Paper:**

The authors propose the use of a learnable hierarchical molecular grammar that can be used for property prediction tasks in the context of drug/materials discovery. This is done by extending the work of Guo et al. (2022) on molecular grammars and adapting it for the supervised learning setting via the usage of a neural diffusion model.

**Summary Of The Review:**

Overall, this is a good paper with a solid theoretical background that deserves credit. I mostly take issue with the (rather limited) result section, which I believe could use further work in terms of tested datasets.

* The authors could consider testing their method on other well-known benchmarking molecular datasets, as available in the MoleculeNet sets to further improve their claims - or elaborate why they chose to test only on the CROW and Permeability datasets.
* The performance of the GEO-DEG approach seems not to be as strong compared to simpler approaches such as D-MPNN when trained on larger datasets (Table 2, Appendix) - to the degree that in the Lipophilicity dataset results are within 1 std. Could the authors elaborate on why they believe this happens?
* A point that seems to be somewhat absent in the study is the computational complexity required to train and evaluate the proposed learned grammar approach and graph diffusion model on new data. Could the authors provide some extra analyses so that the reader gets an idea on how expensive this methodology is compared to other baseline approaches (e.g. D-MPNN)
* In the process for geometry construction, the authors claim that they find it sufficient to use a maximum depth of 10. Could they elaborate  on how that decision was reached?
* The authors motivate hierarchical molecular graphs by claiming that the construction of grammar-induced geometries is costly. In fact, they mention that they find it infeasable to construct grammars when one considers more than ten production rules. Have the authors tried and compare their results with grammars constructed with only a few production rules?
* For PN and the Pre-trained GIN models, how was the pretraining done? Were these trained on additional polymer data or were they pretrained on other molecular species? Additionally on Figure 4b, it is not entirely clear how the Pre-trained GIN baseline was computed. Did the authors use the remaining 80% of the CROW dataset and finetuned the model there? How was this finetuning performed?


Other points:
* What features were used for the training of the random forest baselines?
* Figure 4b could ideally provide additional error bars

---

> ### Author Response · Authors · 2022-11-10
> **Response to Reviewer EFAZ**
>
> **(Q1) Further testing on other benchmarking datasets.**
>
> We have added 4 more benchmarking datasets in our experiments. See **General Reponse** and Appendix A for details. Together with the original experiments, our evaluation includes 8 datasets, 5 of which are widely used benchmarks from MoleculeNet and MUDataset. We show that our method achieves significantly better performance on challenging small regression datasets and outperforms a wide spectrum of baselines on various common benchmarks, including both regression and classification.
>
> **(Q2) Results on Lipophilicity.**
>
> For all the regression tasks in our experiments, we used normalized property values as the training labels. The seemingly marginal difference in the performance of our method on Lipophilicity is due to the fact that the standard deviation of the property values in Lipophilicity is much smaller than that in other datasets. This is also the reason why the performance of the baseline methods on Lipophilicity is much better than that on other datasets. When comparing the direct output error of models (the predicted normalized property values), our method still outperforms all the baselines by a large margin.
>
> **(Q3) Computational complexity analysis.**
>
> Thank you for the suggestion. We have added a detailed computational complexity analysis of our method in Appendix D. Theoretically speaking, the overall computational complexity of our method is the same as that of general graph neural networks in big $\mathcal{O}$ notation. In practice, engineering efforts can be spent on reducing the running time of our method, e.g., parallelization of geometry construction.
>
> **(Q4) Maximum depth of geometry construction.**
>
> We find in practice a 4-degree meta grammar with depth 10 is the minimal depth needed to cover all molecules for all datasets we considered. Increasing the depth will possibly cover more molecular structures, but will also inevitably increase the computational cost of geometry construction.
>
> **(Q5) Comparison with non-hierarchical grammars with only a few production rules.**
>
> In Appendix C, we provide a quantitative comparison of the computational cost of the grammar from [Guo et al. 2022] and the hierarchical grammar we propose. Based on the analysis, we can see our hierarchical grammar greatly reduced the computational cost to the point where the induced geometry can be constructed online, while for non-hierarchical grammars in [Guo et al. 2022], it is usually impossible to construct the geometry even for 5 data points, not to mention a larger training dataset.
> Note that the work [Guo et al. 2022] does not need to construct the geometry for molecular generation.
> Also note that even for 5 data points, in order to cover all the molecule samples, the number of production rules for a non-hierarchical grammar is large.  It is therefore not possible to cover all molecules in any dataset we used with only a few production rules.
>
> **(Q6) Pre-training details for PN and Pre-trained GIN.**
>
> PN itself is a pre-trained model, but we used the GNN features provided by [1] to finetune it on our data. For the pre-trained GIN, we used the pre-trained model and the finetuning code accompanying the original paper [2]. Please note that for our model Geo-GIN, we did not use the pre-trained weights but trained the whole model from scratch on each dataset of our experiments. As you speculated, the horizontal lines in Fig. 4b (and also all the other results of Pre-trained GIN in the tables) were indeed obtained using the full 80% of the remaining data.
>
> **(Q7) Minor Clarifications.**
>
> For the input features of random forest, similar to a common practice in existing works [3, 4], we use Morgan fingerprints implemented by the RDKit package. We also added error bars for Figure 4b.
>
> **(Q8) Code Reproducibility.**
>
> We will publish the implementation of our method as an open-source library upon acceptance and maintain the codebase for future research on molecular grammars.
>
> [1] Stanley et al. Fs-mol: A few-shot learning dataset of
> molecules, In Thirty-fifth Conference on Neural Information Processing Systems Datasets and Benchmarks Track (Round 2), 2021.
>
> [2] Hu et al. Strategies for pre-training graph neural networks, ICLR, 2020.
>
> [3] Capecchi et al. One molecular fingerprint to rule them all: drugs, biomolecules, and the metabolome. J Cheminform, 2020.
>
> [4] Tao et al. Benchmarking machine learning models for polymer informatics: An example of glass transition temperature. Journal of Chemical Information and Modeling, 2021

---

### Official Review · Reviewer_Ahkk · 2022-10-23

**Confidence:** 3
**Correctness:** 3
**Technical Novelty And Significance:** 2
**Empirical Novelty And Significance:** 2
**Recommendation:** 5

**Clarity, Quality, Novelty And Reproducibility:**

Clarity: needs improvement. Need to check referred publications to understand the proposed method and novel contribution of this submission.
Quality: poor. Experimental evaluation is too weak.
Novelty: poor. Incremental improvement of previous methods.
Reproducibility: no code is provided.

**Strength And Weaknesses:**

Strength:

- The proposed method represents grammar production sequences and captures the structure-level similarity between molecules. With a few molecules, this method can utilize the structural relationship between molecules to predict molecular property. It is especially effective for small dataset.

- This paper improves previous molecular hypergraph grammar method [1] by integrating junction tree representation of molecules [2], and proposes a hierarchical molecular grammar to address the computational challenge in construction of grammar-induced geometry.

Weaknesses:

- The proposed method is limited to small dataset scenarios. While there are many pretraining /transfer learning methods work well on both small and large datasets.

- Evaluation is weak. In table 1, the proposed method is only validated on two datasets, compared with limited baseline methods. DILI is a representative small dataset and the authors should at least include experiments on DILI datasets to show the effectiveness of the proposed method, which is claimed to be effective on small datasets. Please refer to [3] for experiments on DILI datasets. Pre-trained GIN is neither the SOTA methods. Please the authors include more high-performance baseline methods. Experiments on large datasets in Table 2 only included two datasets, where there are tens of datasets for evaluation.

- Novelty is limited. The proposed method is an incremental improvement based on previous methods [1, 2].


[1] Guo, Minghao, et al. "Data-efficient graph grammar learning for molecular generation."ICLR (2022.\
[2] Jin, Wengong, Regina Barzilay, and Tommi Jaakkola. "Junction tree variational autoencoder for molecular graph generation." ICML 2018.\
[3] Ma, Hehuan, et al. "Deep graph learning with property augmentation for predicting drug-induced liver injury." Chemical Research in Toxicology 34.2 (2020): 495-506.

**Summary Of The Paper:**

This paper proposes a data-efficient property predictor by utilizing a learnable hierarchical molecular grammar that can generate molecules from grammar production rules. The grammar induces an explicit geometry describing the space of molecular graphs, such that a graph neural diffusion on the geometry can be used to effectively predict property values of molecules on small training datasets. On both small and large datasets, the evaluation shows that this approach outperforms a wide spectrum of baselines, including supervised and pre-trained graph neural networks.

**Summary Of The Review:**

This paper proposes a framework for highly data-efficient property prediction based on a learnable molecular grammar. The intuition behind is that the production rule sequences for molecule generation provide rich information regarding the similarity of molecular structures. The proposed method combines previous existing methods [1, 2] and the novelty is incremental. The major concern is the weak evaluation of the proposed method. I would suggest the authors to include more experimental evaluation.

---

> ### Author Response · Authors · 2022-11-10
> **Response to Reviewer Ahkk**
>
> **(Q1) Extension beyond small dataset scenarios.**
>
> **We believe that our focus on the small datasets is not a limitation but rather, a more general approach than those that work well on only large datasets.** Our primary intention of this paper is to develop a data-efficient property predictor that is capable of **handling extremely small datasets** (with sizes of around 300 or less).
> Data scarcity is a significant bottleneck for many, if not all, deep learning approaches, and this problem is particularly prevalent in many classes of chemistry, including polymers, for example.
> Addressing small dataset scenarios itself is **a non-trivial open research problem** that has been consistently garnering the interest from the field of molecular and material discovery as demonstrated by many recent works [1, 2, 3].
> We have shown in our experiments on CROW and Permeability that our approach can outperform existing strong baselines, including both pre-training learning methods (Pre-trained GIN) and few-shot learning methods (PN) on extremely small datasets.
> This provides considerable empirical evidence that our method is a **substantial improvement** over existing approaches in the area of data-efficient approaches.
>
> To further demonstrate the effectiveness of our method, we have also extended the evaluation of our method on 4 additional benchmarking datasets, see **General Response** and Appendix A for details. Please note that among the datasets, FreeSolv, Lipophiliciy, and ClinTox are relatively large datasets. Our method outperforms other baseline methods on both large and small datasets.
>
> **(Q2) Performance of existing pre-training/transfer learning methods.**
>
> The performance of existing pre-training/transfer learning methods is not comparable to our method, for the following reasons: 1) Pre-training/transfer learning methods mainly focus on classification, a task that is considerably easier than regression, on which we focus extensively; 2) Recent work has shown that even many of these results are not statistically significant and that the challenge of prediction over small data, even in terms of classification, is far from being solved [4]; 3) We demonstrate empirically that pre-training/transfer learning methods suffer from domain gap, whereas our method can be efficiently trained from scratch on small datasets while avoiding this problem.
>
> **(Q3) Evaluations on DILI and more datasets with more baselines.**
>
> In the updated paper, we provide the evaluation results on DILI in Table 4 and also more benchmarking datasets. See **General Reponse** and Appendix A for details.
> Specifically, for ClinTox in Table 5, we compare our approach with a wide range of highly performant baseline methods from a recent paper [5], which by itself proposes a method achieving SOTA on ClinTox.
> Note that these are very strong baseline methods, including those pre-trained using 3D molecular data.
> The overall results show that our method achieves significantly better performance on challenging small regression datasets and outperforms a wide spectrum of baselines on various common benchmarks.
>
> **(Q4) Comparison with more baseline methods.**
>
> Besides comparisons on ClinTox, the baseline methods we experiment with on other datasets are **representatives of highly performant methods** under different categories. For instance, ESAN is one of the recent GNN architectures that achieved SOTA performance across different graph-related tasks. PN is the best performing model for few-shot learning as demonstrated in a recent study [Stanley et al 2021]. HM-GNN is a motif-based GNN model that is one of the most recent and competitive models for molecular property prediction.
> We believe the comparison with these baseline methods is **fair and reasonable** and the results show that our method is a substantial improvement over existing approaches in the area of data-efficient approaches.
> We are also happy to include more baseline methods that the reviewer deems necessary.
>
> **(Q5) Novelty.**
>
> We will post three **General Reponse** threads, separately from this response, to highlight the contribution and significance of this work. We will be happy to answer more questions should any points therein need clarification.
>
> **(Q6) Code Reproducibility.**
>
> We will publish the implementation of our method as an open-source library upon acceptance and maintain the codebase for future research on molecular grammars.
>
> [1] Subramanian et al.  Computational modeling of β-secretase 1 (bace-1) inhibitors using ligand based approaches, Journal of chemical information and modeling, 2016.
>
> [2] Altae-Tran et al. Low data drug discovery with one-shot learning, ACS central science, 2017.
>
> [3] Audus et al. Polymer informatics: Opportunities and challenges, ACS macro letters, 2017.
>
> [4] Wang et al. Evaluating Self-Supervised Learning for Molecular Graph Embeddings, arXiv:2206.08005.
>
> [5] Zhou et al. Uni-Mol: A Universal 3D Molecular Representation Learning Framework, 2022

---

### Official Review · Reviewer_CjiQ · 2022-10-25

**Confidence:** 2
**Correctness:** 2
**Technical Novelty And Significance:** 3
**Empirical Novelty And Significance:** 2
**Recommendation:** 5

**Clarity, Quality, Novelty And Reproducibility:**

The induced grammar part is clear but how to integrate the algorithm in an end-to-end manner is not very clear to me. I hope more clarification can be made and justified to support the claims in the paper.

**Strength And Weaknesses:**

Strength:
1. The molecular grammar hyper-graph construction is innovative.
2. The illustration of grammar-induced geometry is clear and interesting.

Weakness:
1. This paper seems to focus more on grammar tree construction but the detail of how to integrate the prediction model (GRAND) in the paper and induced grammar is not very clear. For example, what if input polymer is not on the induced grammar tree, how do you guarantee every input can be represented on graph.
2. The motivation of using GRAND is not clear. I am assuming it's either because it's performing better or fit the grammar tree. Because the diffusion PDE seems has nothing to do with the proposed framework.
3. The experimental results are not very comprehensive. Admittedly, it shows great performance on two datasets in the paper, however, datasets like ZINC are more frequently used in the literature. Is the proposed method only used for polymers?

**Summary Of The Paper:**

This paper proposes an efficient hierarchical molecular grammar learning algorithm for molecular property prediction. The similar property of same production rule motivates the study of data-efficient grammar-induced geometry. Specifically, the authors describes the process of building hierarchical molecular grammar by pre-defined meta grammar and learnable molecular grammar. Results on both small and large polymer datasets demonstrates the effectiveness of the method.

**Summary Of The Review:**

This paper discusses an efficient training framework for molecular property prediction via grammar-induced geometry. However, I found the motivation and design of the paper is not well supported in the main body of the paper.

---

> ### Author Response · Authors · 2022-11-10
> **Response to Reviewer CijQ**
>
> Thank you for the feedback! Before addressing the detailed comments, please allow us to first go through the overall framework of our approach and summarize how the components are integrated together in an end-to-end manner.
> Given an input set of molecules, our method first constructs a molecular grammar by iteratively sampling the hyperedges in molecular hypergraphs. The learned molecular grammar together with the pre-defined meta grammar can be used to construct a grammar-induced geometry.
> Each molecule is then represented by a graph node in the grammar-induced geometry.
> As part of the diffusion process, each graph node is first initialized with a feature vector, which is then iteratively updated based on information propagated from neighboring graph nodes following the diffusion PDE.
> Finally, property prediction is achieved by transforming the diffused feature vector for each molecule into a scalar value using a linear layer.
> The training process is conducted by minimizing the error between the predicted property values and the ground truth values.
> We refer the reviewer to Appendix K for a graphical illustration of the overall optimization framework and are happy to provide more details if needed.
>
> **(Q1) How to guarantee every input can be represented on graph?**
>
> As described above, the molecular grammar is constructed based on **all input molecules**, i.e., each input molecule will be converted into a junction tree at the end of the grammar construction process. The molecule is then connected to the meta tree node in the meta geometry that represents the isomorphic meta tree to the junction tree. More details can be found in Section 3.2 on Page 6.
>
> **(Q2) Motivation for using GRAND.**
> In our framework, GRAND is a model that exchanges information between molecules in the grammar-induced geometry. GRAND is intended to ensure that molecules lying nearer to one another in the geometry will have more interaction between their features, as they are structurally more similar and can have more similar properties (the key insight of our grammar-induced geometry).
> Ideally, any model that is capable of propagating feature vectors among nodes in the geometry would be acceptable. We choose GRAND since 1) it is widely used to propagate information along geometry domains [1, 2]; 2) GRAND can overcome oversmoothing issues that exist in GNNs as demonstrated in the GRAND paper.
>
>
> **(Q3) More experimental results.**
>
> We have added 4 more benchmarking datasets in our experiments. Please see **General Reponse** and Appendix A for details. Together with the original experiments, our evaluation includes 8 datasets, 5 of which are widely used benchmarks from MoleculeNet and TUDataset. Our method achieves significantly better performance on challenging small regression datasets and outperforms a wide spectrum of baselines on various common benchmarks, including both regression and classification.
>
> **(Q4) "Is the proposed method only used for polymers?"**
>
> Our experiments involve different classes of molecules, including both polymers and small drug-like molecules. See **General Response** and Appendix A for more details. The results demonstrate the effectiveness of our method on both scenarios.
> Yet, we treat polymers as a challenging case in our experiments since the available datasets are much smaller than those for small drug-like molecules. The experiment results on the polymers show that our method is rather data-efficient.
>
> [1] Crane et al, Geodesics in heat: A new approach to computing distance based on heat flow, TOG 2013.
>
> [2] Wang et al, Intrinsic and extrinsic operators for shape analysis, Handbook of Numerical Analysis, 2019.

---

> > ### Comment · Reviewer_CjiQ · 2022-11-18
> > **Thanks for author's response**
> >
> > I have read through author's response. I will re-evaluate my judgement by carefully reading the appendix suggested by the author. One more question, have you tried use other GNN backbone like Graphormer, I still find it less convincing of using GRAND without solid theoretical connections.

---

> > > ### Author Response · Authors · 2022-11-18
> > > **Response to Reviewer CijQ**
> > >
> > > We sincerely thank the reviewer for getting back to us.
> > >
> > > We chose GRAND because it offers experiments on node classification tasks. In these experiments, diffusion is performed on **a single large graph**, and labels are predicted for each node. Our grammar-induced geometry aligns with this setup, where the geometry represents the large graph, with each node representing a molecule whose properties we are attempting to predict. The Graphormer suggested by the reviewer only provides experiments that take **a set of molecular graphs** (much **smaller** in size than our proposed geometry) as input and predict labels for each individual graph. Therefore, we think GRAND represents a better fit.
> > >
> > > We also replaced the GRAND diffusion layer with the Graphormer layer using the [public code](https://github.com/microsoft/Graphormer) and performed experiments on two small regression datasets: CROW and Permeability. We customized Graphormer-SLIM (the smallest one used in the [original paper](https://arxiv.org/pdf/2106.05234.pdf)) to take grammar-induced geometry as input by changing hyperparameters, such as changing hidden dimensions of embedding layers to 300 and modifying the maximal number of nodes/edges, to align with our settings. We followed the original implementation of Graphormer to construct proper inputs for structural encodings to the Graphormer layer. The results compared with our method are as follows (the performance of our method is copied from our draft):
> > >
> > > |                 | CROW | Permeability|#Param.
> > > | ----------- | ----------- | ----------- | ----------- |
> > > Graphormer (MPNN)  | 18.65 | 0.337 | 987,271
> > > Geo-DEG (GIN, GRAND)  |17.0|0.34 |38,530
> > > Geo-DEG (MPNN, GRAND) |18.5 |0.32 |38,530
> > >
> > > Interestingly, Graphomer performs similarly to our method. We attribute this to the number of parameters, as Graphomer layer has far more parameters (including centrality encoding, spatial encoding, edge encoding, and attention layers) compared to GRAND models (which have only one self-attention layer). We believe this is the key reason why Graphormer is competitive in our experiments.
> > >
> > > Altogether, this experiment nicely demonstrates the **flexibility and power** of our framework.

---

### Author Response · Authors · 2022-11-10
**General Response to All Reviewers (1)**


We thank all reviewers very much for the detailed and constructive reviews!
We are grateful that the reviewers found that

1. Our idea is interesting (Reviewer [CjiQ](https://openreview.net/forum?id=SGQi3LgFnqj&noteId=q-0coIOk6V-)), novel (Reviewer [CjiQ](https://openreview.net/forum?id=SGQi3LgFnqj&noteId=q-0coIOk6V-), [EFAZ](https://openreview.net/forum?id=SGQi3LgFnqj&noteId=4qQ89CFCon), [aHZ7](https://openreview.net/forum?id=SGQi3LgFnqj&noteId=Z2FwrpHzpU)) with solid theoretical background (Reviewer [EFAZ](https://openreview.net/forum?id=SGQi3LgFnqj&noteId=4qQ89CFCon)).
2. Our method has the potential to be distinctly impactful to the field (Reviewer [aHZ7](https://openreview.net/forum?id=SGQi3LgFnqj&noteId=Z2FwrpHzpU)).
3. Our experiments have demonstrated the effectiveness of our method in challenging cases involving small datasets  (Reviewer [Ahkk](https://openreview.net/forum?id=SGQi3LgFnqj&noteId=p4urVZwpD-6), [aHZ7](https://openreview.net/forum?id=SGQi3LgFnqj&noteId=Z2FwrpHzpU)).
4. The manuscript provides good level of details with clear description of the model (Reviewer [EFAZ](https://openreview.net/forum?id=SGQi3LgFnqj&noteId=4qQ89CFCon), [aHZ7](https://openreview.net/forum?id=SGQi3LgFnqj&noteId=Z2FwrpHzpU)).


We address common questions and concerns in this overall response and give further details in the individual ones. We are happy to provide additional clarification if needed. Upon reading the reviews, we have made the following changes to the paper (major changes highlighted in red in the paper, updated on Nov. 9th):

1. As all the reviewers are concerned about the evaluation, we have **doubled the number of experiments** in the paper by further including thorough comparisons of 4 additional datasets: HOPV, DILI (as suggested by Reviewer [Ahkk](https://openreview.net/forum?id=SGQi3LgFnqj&noteId=p4urVZwpD-6)), PTC, and ClinTox. Together with the original 4 datasets, our experiments cover: 1) **a wide range of commonly used benchmark datasets** in the literature including MolecuNet (FreeSolv, Lipophilicity, HOPV, ClinTox) and TUDataset (PTC), 2) problems of both **classification** (DILI, PTC, ClinTox) and **regression** (CROW, Permeability, FreeSolv, Lipophilicity, HOPV), as well as 3) datasets of both **small** (CROW, Permeability, HOPV, PTC, DILI) and **large** sizes (FreeSolv, Lipophilicity, ClinTox).

2. We have extended our study for the effect of changing dataset training size on model performance by including the analysis of all 3 small datasets for regression with more baselines compared, as suggested by Reviewers [EFAZ](https://openreview.net/forum?id=SGQi3LgFnqj&noteId=4qQ89CFCon) and [aHZ7](https://openreview.net/forum?id=SGQi3LgFnqj&noteId=Z2FwrpHzpU).

3. We have added a detailed analysis of the cost reduction of geometry construction using the proposed hierarchical molecular grammar as compared to non-hierarchical ones, as suggested by Reviewers [EFAZ](https://openreview.net/forum?id=SGQi3LgFnqj&noteId=4qQ89CFCon) and [aHZ7](https://openreview.net/forum?id=SGQi3LgFnqj&noteId=Z2FwrpHzpU).

4. We have added a computational complexity analysis of the proposed method, as suggested by Reviewer [EFAZ](https://openreview.net/forum?id=SGQi3LgFnqj&noteId=4qQ89CFCon).

5. We have included more references and expanded the elaboration of the related work, as suggested by Reviewer [aHZ7](https://openreview.net/forum?id=SGQi3LgFnqj&noteId=Z2FwrpHzpU).

6. We have incorporated most writing suggestions and figure improvements suggested by reviewers.

-----------------------------------------

In the following parts of this general response, we provide further clarification to address reviewers’ common concerns:

1. Contributions of our work.
2. Technical significance of our proposed grammar-induced geometry.

---

> ### Author Response · Authors · 2022-11-10
> **General Response to All Reviewers (2)**
>
> ## Contributions of our work
>
> While our work shares some high-level similarities with some previous works ([Guo et al 2022] and [Jin et al 2020] as mentioned in the review comments), it **fundamentally differs in both theoretical and empirical insights**, and **should not** be viewed as bare improvement over existing models. Our main contributions are:
>
> **1. We are tackling the extremely small regression task, which is a challenging open research problem in the field and has not yet been well-addressed.**
>
> This paper is primarily intended to develop a data-efficient property predictor that is capable of handling extremely small datasets (with a size of less than 300).
> By all means, **data scarcity** is a common bottleneck for any deep learning approach, and this problem is particularly prevalent in many classes of chemistry, including polymers, for example.
> Many recent studies have demonstrated that handling small dataset scenarios is a **non-trivial open** research problem and is of **practical importance** which has consistently attracted the attention of the fields of molecule and material discovery [1, 2, 3, and more references in Section 1].
> Our model and experiments are both designed to handle this challenging problem, which we believe will lead to new research opportunities in this area.
>
> **2. Our framework, to the best of our knowledge, is the first method that leverages the data efficiency of a learnable molecular grammar for the task of molecular property prediction.**
>
> While the recent renewed trend in using grammars for molecule representation has demonstrated the effectiveness of molecular grammars, existing grammar-based methods (e.g., [Guo et al 2022]) **only focus on molecular generation** since the grammar is a generative model by its nature.
> In terms of property prediction, particularly for extremely small datasets, the task is **significantly more challenging** than molecular generation, as a model needs to learn the mapping from the molecule space to possibly variant property spaces, a much more complicated task than exploring within the molecule space as the generative model does.
> We approach this problem by not trivially deploying a separate property predictor model on top of the grammar, but instead, investigating the underlying reason why most deep-learning methods fail in data-sparse cases and **identifying the key motivation for integrating the grammar** into our framework: **A grammar can provide an explicit prior on structure-level similarity**. The purpose of all developments of our method, including grammar-induced geometry and hierarchical molecular grammar, is to turn this motivation into a practical and effective framework, which is novel and innovative.
>
> **3.** Together with the newly added experiments, **we show that our method achieves significantly better performance on challenging small regression datasets and outperforms a wide spectrum of baselines on various common benchmarks, including both regression and classification.**
>
> **4. We are strong proponents of open research and will open source the implementation and codebase of our method, which can be used as a competitive baseline for future research.**
>
> [1] Subramanian et al.  Computational modeling of β-secretase 1 (bace-1) inhibitors using ligand based approaches, Journal of chemical information and modeling, 2016.
>
> [2] Altae-Tran et al. Low data drug discovery with one-shot learning, ACS central science, 2017.
>
> [3] Audus et al. Polymer informatics: Opportunities and challenges, ACS macro letters, 2017.

---

> > ### Author Response · Authors · 2022-11-10
> > **General Response to All Reviewers (3)**
> >
> > ## Technical significance of our proposed hierarchical molecular grammar and grammar-induced geometry
> >
> > In addition to the contributions, we would like to emphasize the technical significance of our method by highlighting two aspects:
> >
> > **1. Difference from existing molecular grammars (e.g., Guo et al 2022).**
> >
> > Our hierarchical molecular grammar takes the inherent advantages of general molecular grammars, including explicitness, explanatory power, and data efficiency, but **expands the utility and is more compact** than all existing grammars.
> > Existing works only use them for molecular generation; prediction and optimization of molecular properties are accomplished by using a separate model (such as a latent-space model) based on the grammar representation, which is learned separately.
> > We integrate molecular grammar into property prediction tasks by constructing a geometry of the space of molecular graphs, which allows us to **optimize both the grammar and the property predictor simultaneously**.
> > Furthermore, as demonstrated in Appendix C, for non-hierarchical grammars in [Guo et al. 2022], it is usually impossible to construct the geometry even for 5 data points, not to mention a larger training dataset.
> > The hierarchical molecular grammar we propose, **without any loss of expressiveness**, is the first grammar that can be used to construct a geometry **at runtime**.
> > This is achieved by **solid theoretical analysis and guarantee**, which we believe is a significant technical contribution.
> >
> > **2. Difference from existing hierarchical molecular generation (e.g. Jin et al 2018; 2020).**
> >
> > Our approach is fundamentally different from existing hierarchical representations in two respects: 1) **Without the need for any training**, our proposed meta grammar (the coarse level of the grammar) can enumerate all possible junction tree structures by using a compact set of meta production rules, **with theoretical guarantees**.
> > Our method only requires learning at the fine level, which contains molecular fragments determined by the molecular rules, whereas existing methods require learning two models for both coarse and fine levels.
> > 2) As opposed to existing methods that use latent spaces, our meta geometry induced by meta grammar **explicitly models the similarity between molecules** by using graph distance along the geometry. This is achieved by the combined use of a meta grammar, which controls coarse structure similarity, and a fine-level grammar, whose production rules reflect the step-by-step structural evolution of the molecular fragments. The graph distance is also **an explainable measure** of minimal editing distance between molecular graphs, whereas the distance in latent spaces used in existing methods lacks explainability.

---

### Decision · Program_Chairs · 2023-01-20

**Decision:**

Reject

**Justification For Why Not Higher Score:**

The writing needs major work and the empirical contribution is really not clear without comparing to other transductive semi-supervised learning methods.

**Justification For Why Not Lower Score:**

N/A

**Metareview: Summary, Strengths And Weaknesses:**

This work addresses the problem of molecule property prediction when the supervised data is scarce.  It builds on prior work of grammar learning for molecules, which has been previously used for molecule generation. In this work it is used to construct a so-called grammar induced geometry (graph) that describes the generation process of molecules where the leaf nodes represent the molecules, and molecules sharing similar generation process are considered to be similar to one another. To capture this generative similarity, it applies GRAND to the grammar induced geometry to allow information to be exchanged between molecules and the labeled molecules are used to train the decoder and diffusion function of GRAND (as well as the molecular grammar) to perform property prediction on the testing leaf nodes.  Results on both small and large polymer datasets demonstrates the effectiveness of the method.

Pros:
+ The idea of using the grammar to induce a learnable prior for molecule similarities is novel and interesting
+ It appears to work well for property prediction when labeled data is very scarce

Con:
- The writing needs work. The paper spent a lot of space (with redundancy) on the grammar induced geometry, on simple concepts such as BFS, but provided no detail in the main body of the paper on their key technical contribution: how to efficiently construct the geometry and perform molecular grammar learning in an end-to-end manner (the whole thing is deferred to appendix, which is also not self-contained and requires a familiarity with prior work) . It is also unclear how the proposed method can be applied to unseen molecules and ensure that they can be properly represented using the learned grammar (the authors' explanation in their response were similarly opaque and unclear).

- In the experiments presented in the paper, based on their explanation, the proposed method is used in a transductive and semi-supervised manner. But the baselines are all inductive, fully supervised methods. It is not clear the performance gain is not simply due to the transductive and semi-supervised setting.  Comparisons to other transductive and semisupervised methods are missing.



**Summary Of Ac-Reviewer Meeting:**

Discussion is carried out asynchronously via email.